

# Atmospheric Evolution of Molecular Weight Separated Brown Carbon from Biomass Burning

Jenny P.S. Wong[1], Maria Tsagkaraki[2], Irini Tsiodra[2,3], Nikolaos Mihalopoulos[2,4], Kalliopi Violaki[5], Maria Kanakidou[2], Jean Sciare,[6] Athanasios Nenes[1,3,4,7], and Rodney J. Weber[1]

[1]Earth and Atmospheric Sciences, Georgia Institute of Technology, Atlanta, 30331, USA
[2]Environmental Chemical Processes Laboratory, Department of Chemistry, University of Crete, 71003 Heraklion, Greece
[3]ICE-HT, Foundation for Research and Technology Hellas, Patras, 26504, Greece
[4]IERSD, National Observatory of Athens, Palea Penteli, 15236, Greece
[5]Aix-Marseille University, Mediterranean Institute of Oceanography (MIO) UMR 7294, University de Toulon, CNRS, IRD, France
[6]Energy Environment and Water Research Center, The Cyprus Institute, Nicosia 1645, Cyprus
[7]School of Architecture, Civil & Environmental Engineering, École Polytechnique Fédérale de Lausanne, Lausanne, 1015, Switzerland

**Abstract**

Biomass burning is a major source of atmospheric brown carbon (BrC) and through its absorption of UV/VIS radiation, it can play an important role on the planetary radiative balance and atmospheric photochemistry. The considerable uncertainty of BrC impacts is associated with its poorly constrained sources, transformations and atmospheric lifetime. Here we report laboratory experiments that examined changes in the optical properties of the water-soluble BrC fraction of biomass burning particles. Effects of direct UVB photolysis and OH oxidation in the aqueous phase on molecular weight-separated BrC were studied. Results indicated that low molecular weight (MW) BrC (< 400 Da) was rapidly photobleached by both direct photolysis and OH oxidation on an atmospheric timescale of approximately 1 hour. High MW BrC (≥ 400 Da) underwent initial photoenhancement over a few hours, followed by slow photobleaching over ~ ten hours. The laboratory experiments were supported by observations from ambient BrC samples that were collected during the fire seasons in Greece. These samples, containing freshly emitted to aged biomass burning aerosol, were analyzed for both water and methanol soluble BrC. Consistent with the laboratory experiments, high MW BrC dominated the total light absorption at 365 nm for both methanol and water-soluble fractions of ambient samples with atmospheric transport times of 1 to 68 hours. These ambient observations indicate that overall, biomass burning BrC across all molecular weights have an atmospheric lifetime of 15 to 20 hours, consistent with estimates from previous field studies - although the BrC associated with the high MW fraction remains relatively stable and is responsible for light absorption properties of BrC throughout most of its atmospheric lifetime. For ambient samples of aged (> 10 hours) biomass burning emissions, poor linear correlations were found between light absorptivity and levoglucosan, consistent with other studies suggesting a short atmospheric lifetime for levoglucosan. However, a much stronger correlation between light absorptivity and total hydrous sugars was observed, suggesting that they may serve as more robust tracers for aged biomass burning emissions. Overall, the results from this study suggest that robust



model estimates of BrC radiative impacts require consideration of the atmospheric aging of BrC and the stability of high-MW BrC.

## 1 Introduction

Brown Carbon (BrC), the fraction of organic aerosol that absorbs solar radiation in the UV and near-visible wavelengths, may potentially be an important climate warmer with estimated direct radiative forcing that varies from +0.03 to +0.60 W m$^{-2}$ (e.g., Park et al., 2010; Feng et al., 2013; Lin et al., 2014; Wang et al., 2014; Saleh et al., 2015; Jo et al., 2016), and a vertical distribution that can be distinctly different from black carbon and other climate warmers (Zhang et al., 2017). In addition to climate impacts, the importance of UV solar radiation for photochemistry also implies that BrC may affect atmospheric chemistry (He and Carmichael, 1999; Mok et al., 2016).

Current knowledge on the sources, sinks, optical properties and atmospheric lifetime of BrC is limited. Among the sources of BrC, combustion of biomass (Andreae and Gelencsér, 2006; Alexander et al., 2008; Hecobian et al., 2010; Kirchstetter and Thatcher, 2012; Saleh et al., 2014; Lack et al., 2012) and fossil fuels (Bond, 2001; Zhang et al., 2011) are thought to dominate; for example, over one year in SE USA, 50% of BrC was attributed to biomass burning (Hecobian et al., 2010). Secondary processes that often involve carbonyls and nitrogen-containing compounds can also generate BrC in the atmosphere (Laskin et al., 2015). Molecular identification of the chromophores is a challenging task, as there may be a multitude of light absorbing compounds. It remains unclear whether BrC is comprised of low concentrations of strongly absorbing chromophores, or a large number of weakly-absorbing chromophores in a complex organic matrix. To date, several classes of compounds have been identified as BrC in biomass burning organic aerosols (BBOA), such as nitroaromatic compounds (Desyaterik et al., 2013; Zhang et al., 2013; Mohr et al., 2013; Lin et al., 2016), humic-like substances (HULIS; Dinar et al., 2008; Hoffer et al., 2006; Fan et al., 2016; Wang et al., 2017), and other high molecular weight substances (i.e., compounds > 400 Da) (Di Lorenzo and Young, 2016; Di Lorenzo et al., 2017; Wong et al., 2017).

Studies have suggested that following the emission or formation of biomass burning BrC, their optical properties can be transformed by atmospheric aging processes. Laboratory studies, for both model compounds and complex mixtures of biomass burning BrC, suggested that initial stages of photochemical aging can increase light absorption ("photoenhancement"), followed by a subsequent decrease ("photobleaching") (Saleh et al., 2013; Zhong and Jang, 2014; Zhao et al., 2015; Sumlin et al., 2017; Hems and Abbatt, 2018). The atmospheric lifetime was constrained only for biomass burning BrC from nitrophenols, where photobleaching by aqueous OH oxidation in fog or cloud droplets was found to be their dominant atmospheric loss mechanism, with an estimated atmospheric lifetime of a few hours (Zhao et al., 2015; Hems and Abbatt, 2018). Field observations from wildfire emissions in the Northwestern USA and Amazon have suggested that most biomass burning BrC chromophores have an atmospheric lifetime in the range of 13 to 30 hours (Forrister et al., 2015; Wang et al., 2016). It was observed, however, that the majority of light absorption in aged biomass burning BrC (approximately 2 days of atmospheric transport) from wildfire emissions in Northeastern Canada, was associated with compounds of 500 Da and larger (Di Lorenzo



and Young, 2016; Di Lorenzo et al., 2017). Altogether, this indicated that a fraction of high molecular weight BrC is recalcitrant to atmospheric aging processes. While there is field evidence that aqueous-phase chemistry can transform the optical properties of BrC emitted from biomass burning (Gilardoni et al., 2016; Zhang et al., 2017), the specific aging processes leading to these field observations remain unknown. In addition, these contrasting laboratory and field observational constraints on the atmospheric lifetime of BrC are much shorter than the assumed atmospheric lifetime of BrC (approximately 4 days) utilized in models to estimate its impacts on aerosol direct radiative forcing (Jo et al., 2016).

Along with constraining the atmospheric lifetime of biomass burning BrC, accurate estimates of the contribution of biomass burning to global BrC are also critical for robust assessments of its climatic impacts. Levoglucosan, an anhydrous sugar emitted during biomass pyrolysis, is a molecular tracer widely used to estimate the contribution of biomass burning to ambient organic aerosol concentrations, as it was historically thought to be chemically inert (Simoneit et al., 1999). Yet, a growing number of studies have demonstrated that levoglucosan is subject to significant atmospheric loss, with an estimated atmospheric lifetime of 0.7 to 2.2 days (Hennigan et al., 2010; Hoffmann et al., 2010; Kessler et al., 2010; May et al., 2012; Slade and Knopf, 2014; Zhao et al., 2014; Sang et al., 2016). Given that the average lifetime of atmospheric aerosols with respect to deposition is considerably longer, and that recent field observations have demonstrated aged BBOA had negligible concentrations of levoglucosan (Bougiatioti et al., 2014; Zhou et al., 2017; Theodosi et al., 2018), these results suggest levoglucosan cannot be used to estimate BrC levels in aged biomass burning emissions (> 1 day).

In this study, we systematically investigated the photochemical aging of molecular weight-separated water-soluble (WS) BrC fraction from biomass burning by both aqueous OH oxidation and photolysis by UVB radiation. We build upon earlier laboratory experiments that examined the aging effects of photolysis by UVA radiation (Wong et al., 2017) to establish a more complete understanding of the effects of different photochemical aging processes. Based on the photobleaching rates determined from these laboratory experiments, we estimated the dominant photochemical pathway leading to the loss of biomass burning BrC in the atmosphere. To provide additional field evidence of the impacts of atmospheric aging on biomass burning BrC, and to further assess the atmospheric stability of high molecular weight BrC, the light absorptivity of molecular weight-separated water and methanol (MeOH) soluble BrC fraction from ambient samples of different atmospheric ages was determined. Finally, from these ambient samples, we assessed the use of levoglucosan and other proposed biomass burning species as robust tracers for aged biomass burning aerosol and BrC.

## 2 Experimental

### 2.1 Laboratory Experiments

#### 2.1.1 Preparation of WS BrC

Wood smoke BrC was generated in the laboratory using the method described in Wong et al. (2017). Briefly, a small piece of dry cherry hardwood (5 - 10g), placed on the bottom of a cylindrical electronically-heated combustor, was pyrolyzed



under an oxygen-free atmosphere at 210°C. The resulting smoke stream was subsequently diluted with filtered air by a factor of ~3 and the BBOA was collected on polytetrafluoroethylene filters (47 mm, 2µm pore size, Pall Corporation) at 6 lpm for 100 min and stored at -10°C. Prior to each laboratory experiment, water-soluble (WS) BrC was extracted from the filter by adding 15 mL of purified water (18.2 mΩ) in a sealed glass vial and sonicated for 60 min. The extract was filtered using a 0.2 µm PTFE syringe filter (Fisher) to remove any insoluble materials that may damage the chromatography column or plug the waveguide used to characterize BrC properties (Section 2.1.4). Given that the WS BrC is dissolved in bulk aqueous solutions, the experimental conditions for the photochemical aging experiments, such as BrC concentrations and viscosity, most likely represent the aging of WS wood smoke BrC in fog and/or cloud droplets. For these laboratory studies, we only focused on the aging of WS BrC, as results from our previous work indicated that the majority of the light absorption of laboratory generated BrC from wood smoke was contributed by WS fraction (~ 77%).

### 2.1.2 Photolysis of WS BrC by UVB lights

Experiments examining the photolytic aging of WS BrC by UVB lights follow the same experimental procedure for UVA photolysis as described in Wong et al. (2017). All photochemical aging experiments were conducted in a photoreactor, with a slowly rotating vial rack (40 rpm) placed in the center and surrounded by 16 UVB lamps (Desert Series 50 T8, Zilla). With all UV lamps on, continuous ventilation by two fans maintained the temperature inside the photoreactor to 30 ± 1°C. The photon fluxes inside the photoreactor from UVB lamps were determined by chemical actinometry using 2-nitrobenzaldehyde (see Wong et al., 2017 for experimental details). The wavelength dependent photon fluxes from both UVA and UVB lamps are shown in Figure S1, where the actinic flux at solar noon is provided for comparison. Most of the radiation fell in the range of 300 – 400 nm, with a maximum at 355 nm and 310 nm for UVA and UVB lamps, respectively. Note that while both UVB and UVA lamps have comparably similar photon fluxes for wavelengths lower than 320nm, the UVA lamps have correspondingly much higher photon flux at higher wavelengths. These spectral differences allowed for investigating the wavelength dependence of BrC aging by photolysis.

For each photolysis experiment, multiple, pre-cleaned 2 mL borosilicate glass vials (sealed with Teflon-lined caps), each containing 1mL of the WS BrC extract solution, were placed on the rotating vial rack inside the photoreactor. Each extract solution was diluted by 5% (with purified water) such that the final concentration of WS BrC used in the photolysis experiments are identical to those used in the aqueous OH oxidation experiments, where the WS BrC concentration was diluted by 5% from the addition of $H_2O_2$ (Section 2.1.3). The vials were illuminated by UVB lights for up to approximately 100 hours, where at different illumination times, one vial was removed for offline analysis (described in Sec. 2.1.4). Control experiments were conducted; no changes in WS BrC properties were observed when the vials containing the extract were completely covered by aluminium foil, where they were only exposed to the elevated temperatures inside the photoreactor, and not UVB radiation. Photolysis experiments were repeated at least 3 times to ensure reproducibility.





### 2.1.3 Aqueous OH Oxidation of WS BrC

Experiments for aqueous OH oxidation were conducted in the same experimental setup used in the photolysis experiments. $H_2O_2$ (30%, Sigma Aldrich) was added to the WS BrC extract solution (final concentration of 1.5 mM) as a photolytic source of OH radical upon irradiation with UVB lights (up to 18 hours). Dark control experiments were conducted

to confirm that the dark reaction of $H_2O_2$ with WS BrC did not change its optical or molecular weight properties. Similar to photolysis experiments, at different illumination times, one vial was removed for offline analysis. The OH oxidation experiments were repeated 3 times to ensure reproducibility.

The steady-state OH concentration ($[OH]_{ss}$) in these photochemical oxidation experiments was determined in order to estimate the atmospheric lifetime of WS BrC with respect to aqueous OH oxidation. This was performed by monitoring the

formation of para-hydroxybenzoic acid (*p*-HBA) from the reaction of OH with benzoate, a commonly employed OH scanvenger (Zhou and Mopper, 1990; Anastasio and McGregor, 2001; Badali et al., 2015). The OH quantification experiments are described in detail in Section S1. Briefly, sodium benzoate (0.1 – 1.0 µM) was added to WS wood smoke BrC extract solution with and without 1.5mM $H_2O_2$. The resulting solutions were illuminated for up to 18 hours, where at different times, a sample vial was removed to determine the concentration of *p*-HBA, using HPLC-UV/VIS absorption. The yield of *p*-HBA

from the reaction of OH with benzoate (0.17) was used to convert the formation rate of *p*-HBA to an OH production rate, from which the $[OH]_{ss}$ was estimated. *p*-HBA does not absorb radiation in the same wavelength regions as BrC, but given that other products formed from the reaction of OH + benzoate do, in order to avoid measurement interference from these products, the OH quantification experiments were conducted separately from the experiments where the effects of OH oxidation on BrC properties were examined. In these experiments, the relationship between OH photo-production and benzoate concentration

was determined in order to quantify $[OH]_{ss}$ in the aqueous OH oxidation experiments where the BrC properties were monitored (i.e., no benzoate as an added OH scavenger). Figure S2 shows the relationship between the *p*-HBA formation rate and concentration of added benzoate was used to estimate the $[OH]_{ss}$ when [benzoate] = 0 (i.e., experiments where light absorptivity of BrC was monitored).

### 2.1.4 Offline WS BrC Measurements

Following the removal of each sample vial from the photoreactor, the BrC solutions were divided to determine various BrC chemical and optical properties, using the same procedures outlined in Wong et al. (2017). Briefly, changes in the water-soluble organic carbon (WSOC) concentration due to photochemical aging were monitored using a Sievers Total Organic Carbon (TOC) Analyzer (Model 900, GE Analytical Instruments). For these measurements, the WS BrC samples were diluted by a factor of 250 to ensure the WSOC concentrations were in the linear range of the instrument, which was routinely calibrated

using solutions of dissolved sucrose of known concentrations. Prior to photochemical aging, each sample vial contained 1400 $\pm$ 178 ppb of WSOC. The light absorptivity of WS BrC of all molecular weights (i.e., bulk WS BrC) was monitored using the 250x diluted WS BrC solutions and an absorption spectrometer consisting of a liquid waveguide capillary (2.5 m optical path-



length, World Precision), a deuterium tungsten halogen light source (DT-Mini, Ocean Optics), and a light detector (USB4000, Ocean Optics) that can continuously monitor all wavelengths between 230 and 800 nm. The molecular weight distributions of WS BrC were determined using high-performance liquid chromatography (HPLC; GP40 Dionex), equipped with a size exclusion chromatography (SEC) column (Polysep GFC P-3000, Phenomenex) that was operated in isocratic mode using a

90:10 $v/v$ mixture of water and methanol with 25 mM ammonium acetate as the mobile phase, at 1mL min$^{-1}$. The near-UV/VIS absorbance of the molecular-weight separated BrC compounds were monitored using an absorbance spectrometer that was coupled in-line with the SEC system. The spectrometer consisted of the same components as the one used for bulk WS BrC measurements, except a liquid waveguide capillary with a 1 m optical path-length (World Precision) and a different model of the deuterium tungsten halogen light source (DT-Mini-2B, Ocean Optics) were used. This SEC approach was routinely

calibrated using standards of known molecular weights, of which the calibration methodology and  the relationship between molecular weights and elution volumes were previously described (Wong et al., 2017). We note that the molecular weights reported using this SEC approach are only approximate, as the accuracy of the molecular weight calibration depends on whether the molecular densities of calibration standards are representative of that of WS BrC molecules, which are currently unknown.

## 2.2 Field Observations on Crete Island

### 2.2.1. Sampling Site and Identification of Biomass Burning Events

Filter samples containing ambient BrC from biomass burning emissions were collected during the fire season (July to October) of 2016 and 2017 at the Heraklion station on Crete, Greece (Figure 1). During the fire season, persistent northerly winds (the Etesians) move air masses across the Aegean Sea, where no further contribution from fire emissions can occur,

transporting biomass burning emissions from continental Eastern Europe towards the sampling site. The fire season in 2017 was more intense compared to 2016, due to extended droughts and high temperatures.

Information on location and date of fire events, along with fire radiative power were obtained from the Fire Information for Resource Management System (FIRES, http://firms.modaps.eosdis.nasa.gov/firemap), which was detected by the Moderate Resolution Imaging Spectroradiometer (MODIS). Fire radiative power (in megawatts) was used as a rough proxy

for biomass burning emission rate (Wooster, 2002). Fires occurring 3 days prior and within each filter sampling period, with fire radiative power over 100 megawatts, were included in the analysis. Airmass back trajectories were computed via HYSPLIT (Stein et al., 2015), using archived Global Dara Assimilation System (GDAS) meteorology, from which the vertical velocity was determined. New trajectories were computed every hour of each filter sampling interval (i.e., 22 – 24 hours), for a total run time of 72 hours. The information from FIRES and back trajectories were used together to identify filters samples were

influenced by fire events (i.e., intersection of back trajectories with the fire locations) and to estimate the corresponding atmospheric transport time from the fire location to the sampling site (example is shown in Figure S3). This analysis approach was chosen to explore the stability of biomass burning tracers, however, it does not account for variability in BrC emissions



from various fires. Out of 65 field filter samples collected in the 2016 and 2017 fire seasons (method discussed below), 24 biomass burning filter samples of various atmospheric transport times were identified. We focus our following analysis on these identified biomass burning events, shown in Figure 1. Note that several fires occurring near Athens and on Chios Island were burn events that persisted for multiple days.

## 2.2.2 Filter Collection, Extraction and Analysis

Ambient BrC in $PM_{2.5}$ were collected on pre-baked $8 \times 10$ in. quartz filters (2500QAT-UP, Pall) using a high-volume (Hi-Vol) sampler (TISCH) for 22 – 24 hours at a flow rate of 1.4 and 2 $m^3$ $min^{-1}$ for 2016 and 2017, respectively. Immediately after collection, the filter samples were wrapped in prebaked aluminium foil and stored at -10°C until analysis. Online black carbon absorption measurements were conducted during the fire season of 2017, using a seven wavelength aethalometer (AE31, Magee Scientific).  In addition, multiple field blanks were collected during both fire seasons. Each quartz filter sample was divided into portions for the determination of various chemical properties. A 1.5 $cm^2$ punch of the filter sample was analyzed for organic carbon and elemental carbon using an OCEC Analyzer (Sunset Laboratory Inc) using the NIOSH Method. Another 1.5 $cm^2$ filter punch was used to determine water-soluble components after extraction by sonication. This included analysis of carbohydrates (levoglucosan, mannosan, galactosan, glucose, mannose, and galactose) using high performance anion exchange chromatography with pulsed amperometric detection (HPAEC-PAD; ICS-3000, Dionex), which is described in Fourtzio et al. (2017). Using a separate ion chromatographic system (Dionex), analysis of anions ($Cl^-$, $Br^-$, $NO_3^-$, $SO_4^{2-}$, $C_2O_4^{2-}$; CS12A column with CERS 500 suppressor) and cations ($K^+$, $Na^+$, $Ca^{2+}$, $Mg^{2+}$, $NH_4^+$; AS4A-SC column with AERS 500 suppressor) were also conducted. Potassium associated with biomass burning (i.e., non-sea salt $K^+$) was determined from the total potassium minus potassium associated with sea-salt. This was calculated using the measured $Na^+$ concentrations and a standard seawater $K^+$-to-$Na^+$ mass ratio of 0.0359 (Seinfeld and Pandis, 1998).

Two 1.5 $cm^2$ punches were placed in a pre-cleaned 2mL borosilicate glass vial where either 1 mL of purified water (18.2 mΩ) or methanol (MeOH; HPLC grade, Merck) was added and sonicated for 1 hour to extract either the water soluble (WS) or MeOH soluble (i.e., water-soluble and water-insoluble) BrC, respectively. Note that the extractions of MeOH BrC and WS BrC were done on separate sections of the same filter. Each extract was then filtered using a new 0.2 µm PTFE syringe filter (Fisher). For WS BrC, an aliquot of the filtered extract was used to determine the WSOC concentration using the TOC instrument discussed in Section. 2.1.4. For both WS- and MeOH BrC filtered extracts, aliquots of the solutions were used to determine the molecular weight distributions of BrC, using the HPLC-SEC-UV/VIS absorption technique discussed in Section 2.1.4. Additionally, an aliquot of both filtered solutions was used to measure the bulk light absorption properties (i.e., not molecular weight separated) of WS and MeOH BrC, using the absorption spectrometer described in Section 2.1.4. Note that for both TOC and bulk UV/VIS absorption measurements, the filtered extracts were diluted by a factor of 20 to ensure the measured properties were in the linear response range of the instruments. The field filter blanks were analysed using the same methodology as the BBOA filters and all measurements were blank subtracted.

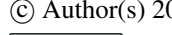



## 3 Results and Discussion

### 3.1 Laboratory Experiments on WS BrC

#### 3.1.1 Bulk WS BrC

Upon illumination by UVB lights, losses in WSOC were observed for both direct UVB photolysis and UVB + $H_2O_2$

experiments, as shown in the blue and green traces of Figure 2a. The effects of aqueous OH oxidation is taken to be the

difference between the UVB and UVB + $H_2O_2$ experiments, assuming that the effects of direct UVB photolysis are identical

in both types of experiments (i.e., the addition of $H_2O_2$ did not significantly alter the rate of direct photolysis). Here, aqueous

OH oxidation did not lead to additional lost of WSOC, indicating that the loss of WSOC is only due to direct UVB photolysis.

The effects of photochemical aging on the light absorption per water-soluble organic carbon (mass absorption

coefficient, MAC) of WS BrC are shown on Figure 2b. The calculation method for the MAC at 365 and 400 nm is discussed

previously in Wong et al., (2017). Initial increase in MAC values were observed due to photochemical aging, suggesting that

the WS BrC undergo photoenhancement, leading to increased absorptivity of radiation at 365 and 400 nm. Given that a loss

in WSOC was observed during this photoenhancement period, the increased MAC values may be driven by a loss in non-

absorbing WSOC and/or the formation of more absorbing WS BrC. The changes in WSOC and $MAC_{365}$ are similar for UVB

and UVB + $H_2O_2$ experiments, suggesting that the aging by UVB led to the observed photoenhancement. Following this period

of initial photoenhancement (up to 12 hours), photobleaching (i.e., decrease in MAC values) of WS BrC was observed. Here,

a steeper slope for the decrease in MAC values were observed for the UVB + $H_2O_2$ compared to the UVB experiments,

indicating that aqueous OH oxidation leads to enhanced decay in light absorptivity, and that while BrC are susceptible to both

degradation due to UVB photolysis and OH oxidation, certain BrC chromophores are more reactive towards OH radicals.

Previous studies that have examined the photochemical aging of model biomass burning aromatic compounds (Gelencsér et

al., 2003; Chang and Thompson, 2010; Ofner et al., 2011; Zhao et al., 2015; Smith et al., 2016; Hems and Abbatt, 2018),

surrogate mixtures of biomass burning BrC (Schnitzler and Abbatt, 2018), and BrC emitted from the pyrolysis of various types

of biomass (Zhao et al., 2015; Wong et al., 2017; Sumlin et al., 2017) have similarly observed initial photoenhancement,

followed by photobleaching. It has been proposed that polymerization/functionalization of BrC leads to photoenhancement,

while fragmentation results in photobleaching.

#### 3.1.2 WS BrC separated by Molecular Weight

Molecular weight separated BrC measurements by SEC provide additional insight into the reactivity of specific

classes of WS BrC molecules leading to the observed bulk photoenhancement and photobleaching. This is illustrated in Figure

3 and Figure 4, where the $Abs_{365}$ were binned according to elution volumes, where the high-molecular weight fraction (high-

30    MW) is defined as the sum of light absorptivity ($Abs_{365}$) for molecules with approximate molecular weights between 66K Da

and 401 Da (i.e., elution volumes between 8 and 15 mL) and low molecular weight fraction (low-MW) as the sum of light

absorptivity ($Abs_{365}$) for molecules with approximate molecular weights of 400 Da and less (i.e., elution volumes higher than





15 mL). Prior to photochemical aging, fresh WS BrC consisted of both high- and low-MW chromophores (Figure 3a), however, the contribution of these two fractions to light absorption change when photochemically aged. Figure 4 shows that both molecular weight fractions exhibit dynamic changes in their light absorptivity due to photochemical aging, but to a different extent. High-MW WS BrC undergo substantial intial photoenhancement, followed by photobleaching. Low-MW WS

BrC initially decayed within the first hour, then there was an increase in light absorptivity between 1 and 5 hours of illumination time, followed by another period of decreasing light absorptivity. The two separate phases of decreasing light absorptivity suggest that low-MW WS BrC contains chromophores of different photoreactivity. While the photobleaching of both molecular weight fractions appeared largely to follow pseudo first-order kinetics, the photoenhancement of high-MW WS BrC exhibited non-first order behavior. This may be due to the presence of chromophores with different reactivity, or that after the

initial period of photoenhancement, further aging do not lead to further increases in light absorptivity. For simplicity, we treat this period of photoenhancement as pseudo first-order. In order to isolate the effects of OH oxidation on light absorptivity, we follow the approach used by Zhao et al. (2015), where the pseudo first-order OH oxidation rate constants ($k^I_{OH}$) of photoenhancement/photobleaching were calculated by taking the difference between the observed first-order growth/decay rate constants due to UVB photolysis ($k^I_{UVB}$) and UVB + $H_2O_2$ photolysis experiments ($k^I_{UVB+H2O2}$), using Eq. (1). The

second-order OH oxidation rate constant ($k^{II}_{OH}$) can be calculated from Eq. (2).

$$k^I_{OH} = k^I_{UVB+H2O2} - k^I_{UVB} \qquad (1)$$

$$k^{II}_{OH} = k^{II}_{OH} / [OH]_{ss} \qquad (2)$$

We note that for Eq. (2), $[OH]_{ss}$ represents the steady state conentration of OH radicals due to the photolysis of 1.5 mM of $H_2O_2$ (i.e., difference in $[OH]_{ss}$ between UVB photolysis and UVB+ $H_2O_2$ photolysis experiments). Additionally, for the high-

MW BrC, the photoenhancement and photobleaching rate constants were determined by fitting first-order curves to the first 3 hours and between 8 and 18 hours of absorption data at 365nm (Figure S4a). For the low-MW WS BrC, to determine the decay rate constants of rapidly and slowly photobleached chromophores, a first-order decay curve was fitted to 0 to 1 hours and 8 to 18 hours of absorption data at 365 nm, respectively (Figure S4b).

The resulting observed rate constants for photoenhancement and photobleaching due to aqueous OH oxidation and

UVB photolysis are shown in Table 1. Here, the corresponding rate constants due to UVA photolysis, which were determined by our previous work (Wong et al., 2017) are included for comparison. Considering the reaction rate constants for low-MW WS BrC, their intial photoenhancment was only observed in the presence of UVA radiation (i.e., enhanced photon flux at wavelengths above 310 nm), indicating reactions leading to the increased light absorptivity are wavelength dependent. Whereas photochemical aging of low-MW WS BrC by UVB photolysis and aqueous OH oxidation only leads to

photobleaching. As mentioned previously, low-MW WS BrC of different reactivities with respect to photobleaching by aqueous OH oxidation and UVB photolysis were observed; chromophores that were rapidly photobleached due to exposure to UVB lights and chromophores that were slowly photobleached by both aqueous OH oxidiation and UVB radiation. We note that the second-order rate constant for the decay in light absorptivity due to the OH reaction with slowly photobleached low-MW WS chromophores determined in this study [$(2.9 \pm 0.5) \times 10^9$ $M^{-1}$ $s^{-1}$] is comparable to the range of the concentration-



based rate constants for the OH reaction with 3 different nitrophenols ([(3.7 - 5.0) × $10^9$ $M^{-1}$ $s^{-1}$], as reported by Hems and Abbatt (2017). Given that nitrophenols, a class of WS BrC that have been detected in BBOA in significant concentrations (Mohr et al., 2013; Lin et al., 2016), and have molecular weights that are approximately less than 200 Da, it is reasonable that they govern the OH reactivity of low-MW WS BrC (i.e., ≤ 400 Da) observed in this study.

5       Considering the evolution of high-MW WS BrC (Figure 4a), at early reaction times, changes in their light absorptivity were only observed due to UVB exposure, where there was only initial photoenhancement (up to 15 hours), followed by photobleaching. Given that the photoenhancement and photobleaching rates for high-MW WS chromophores due to UVB and UVA exposure (Table 1) are similar, and that both UVB and UVA lamps have similar photon fluxes for wavelengths lower than 310nm, our results suggest that most of the photochemical aging was initiated by UVB radiation. Owing to their rapid

photoenhancement and slow photobleaching, the contribution of high-MW WS BrC to total light absorptivity increases throughout photochemical aging, from 20% when the WS BrC was freshly emitted, up to 80% after 100 hours of UV exposure (Figure 4b). Given that the integrated UVB photon flux in these laboratory experiments is roughly 92% of the sun at Solar Noon (i.e., one hour of UVB exposure in laboratory experiments is equivalent to 0.92 hours in the atmosphere), these results further support earlier field observations where the light absorptivity of aged (up to 40 hours) ambient biomass burning WS

BrC was attributed to molecules larger than 500 Da (Di Lorenzo and Young, 2016; Di Lorenzo et al., 2017).

      Changes in the absorption Angström exponent (AAE) throughout UVB photolysis and OH oxidation were also determined from linear regression fits to logAbs vs. logλ (Figure S5) in the wavelength ranges of 320-500 nm for both high and low-MW fractions of WS BrC molecules, and 320-420 nm for high-MW BrC. This was done to provide additional insight into the effects of photochemical aging on the light absorption spectral properties of BrC (Figure 5). Prior to any photochemical

aging, different AAE values for low-MW (10.0 ± 0.4) and high-MW (6.4 ± 0.7) WS BrC were observed, indicating that the low-MW WS BrC have a much stronger spectral dependence than that of high-MW WS BrC. We speculate that the lower AAE values of high-MW WS BrC are due to the highly conjugated nature of these molecules, as previous observations by Hopkins et al. (2007) have indicated that biomass burning BrC with lower AAE values have a higher extent of carbon $sp^2$ hybridization compared to those with higher AAE values. As the low- and high-MW WS BrC were photochemically aged,

changes in their AAE values were observed. For low-MW WS BrC, AAE values decreased due to decreased light absorptivity at lower wavelengths. For high-MW WS BrC, comparison of the temporal evolution of the AAE values determined in both wavelength ranges (Figure5b-c) shows that during initial photoenhancement (up to 15 hours), a slight decrease in AAE was only observed from 320-420nm, suggesting that photoenhancement reactions, such as functionalization and polymerization, enhance the absorptivity of light of wavelengths only up to 500 nm. Following this period of initial photoenhancement,

increasing AAE values were observed as the high-MW WS BrC was photobleached, where the increase in AAE values for 320-500 nm was more rapid compared to 320-420nm, indicating a blue shift in the absorption spectra of high-MW WS BrC due to decreasing absorption in the 420 to 500 nm range. Collectively, the differences in evolution of AAE values further support that the reactivity and aging mechanisms of BrC are dependent on their molecular weight.



### 3.1.3 Atmospheric Fate of WS BrC from Biomass Burning

These experimentally determined rate constants indicate that ambient WS BrC are transformed within a day in the atmosphere (Table S1; calculation method discussed in Section S2), and considering that the average atmospheric lifetime of particles with respect to deposition is approximately one week, our laboratory results indicate that photochemical aging has an important effect on the optical properties of BBOA. For low-MW WS BrC, laboratory results suggest that there are two groups with different reactivities. A highly photolabile fraction of low-MW WS BrC is photobleached in the atmosphere, with UVB photolysis being their dominant atmospheric fate with an estimated atmospheric lifetime of $0.7 \pm 0.2$ hours. A second, less reactive fraction of low-MW WS BrC have an estimated atmospheric lifetime of $9.8 \pm 1.6$ hours, with aqueous OH oxidation representing its dominant atmospheric loss mechanism. However, this less reactive low-MW WS BrC does not significantly contribute to total light absorptivity, as their decay only increased the fractional contribution of high-MW to total light absorptivity for WS BrC by 0.2 (Figure 4b). Conversely, for high-MW WS BrC, following their initial photoenhancement where color formation occurs over a timescale of a few hours, these BrC are photobleached by UVB photolysis, with atmospheric lifetime of $11 \pm 2.3$ hours.

While these estimated atmospheric lifetimes suggest that the majority of BrC are photobleached in the atmosphere, it is important to note that for high-MW WS BrC, the rate of decreasing light absorptivity slows down with time, where after 100 hours of photochemical aging in the laboratory, approximately 20% of the initial light absorptivity remained (Figure 4a). These observations suggest that a fraction of high-MW WS-BrC have an atmospheric lifetime longer than 11 hours and are more persistent. We note that the estimated atmospheric lifetimes were calculated assuming continuous exposure to solar radiation with an actinic flux corresponding to that at Solar Noon and $[OH]_{ss}$ of $1.0 \times 10^{-14}$ M, which represent the daily peak solar photon flux and the upper range of OH concentrations in cloud droplets (Herrmann et al., 2010; Arakaki et al., 2013), thus likely representing the lower range of BrC atmospheric lifetimes. We also stress that there are uncertainities in these estimates, as they assume that the photolysis quantum yield of BrC is wavelength independent. Also, estimates of $[OH]_{ss}$ in the condensed phase, which includes aqueous particles, cloud and fog droplets, range over several orders of magnitude (Arakaki et al., 2013; Ervens, 2015). Under lower levels of $[OH]_{ss}$, which better represents the oxidant concentrations of aerosol particles, photolysis by UVB becomes the domiant atmospheric loss mechanism for all molecular weight fractions of BrC.

### 3.2 Field Observations

### 3.2.1 Bulk and Molecular Weight Separated WS and MeOH BrC

Investigating the impacts of atmospheric aging on biomass burning BrC in the field and comparison to laboratory observations represent the other primary goals of this study. Both WS and MeOH (i.e., water-soluble and insoluble) BrC were analysed for the field samples, to assess the atmospheric evolution of BrC from ambient BBOA. Evolution of various properties of WS and MeOH BrC as a function of atmospheric transport time is shown in Figure 6, where all individuals measurements





(filled points), along with binned (every 5 hours) median values (open points) are presented. We first discuss the bulk measurements (Figure 6a-d).

Considering field samples with atmospheric transport times up to 10 hours, an increase in bulk (i.e., not molecular weight separated) $Abs_{365nm}$ (Figure 6b) was observed for both MeOH and WS BrC, while a significant increase in $MAC_{365}$
values (Figure 6c) was only observed for the water-soluble portion. This is due to the more significant loss of WSOC compared to OC (Figure 6a), suggesting that the non-absorbing water-soluble compounds in biomass burning are more rapidly lost in the atmosphere due to aging. For field samples with atmospheric transport times longer than 10 hours, lower values of WSOC, OC, $Abs_{365}$ values for MeOH and WS BrC were observed, suggesting that atmospheric aging processes led to a decay in the mass and light absorptivity of biomass burning BrC. These ambient observations of bulk WS BrC corroborate our earlier
laboratory results, suggesting that atmospheric photochemical aging processes increase MAC values for WS BrC, at least in fresher biomass burning plumes, followed by photobleaching with further atmospheric aging. Despite the considerable scatter, the field data indicate that the light absorptivity of bulk MeOH BrC decreased at a slower rate compared to bulk WS BrC (see Figure S6 for exponential decay). From these exponential decays in bulk MeOH and WS ambient BrC, the atmospheric lifetime of bulk MeOH and WS BrC is estimated to be between 15 – 19 hours.

Lastly, from these bulk measurements, AAE values did not change significantly throughout atmospheric transport, suggesting that aging processes due not greatly affect the wavelength dependence of BrC light absorption. We note that while black carbon mass concentrations were only measured at the field site during the 2017 fire season, AAE values were observed to be inversely proportional to the black carbon to organic aerosol ratios (Figure S7), further extending previous demonstrations of this relationship from laboratory-generated and ambient biomass burning BrC (Saleh et al., 2014; Lu et al., 2015; Gilardoni
et al., 2016; Costabile et al., 2017).

Molecular weight separated BrC measurements from the corresponding field samples (Figure 6e) indicate that even for fresh biomass burning emissions (i.e., one hour of atmospheric transport time), high-MW MeOH and WS BrC dominated the total light absorption at 365 nm. This is also highlighted in Figure 3b, which shows the molecular weight separated absorption spectra of fresh ambient biomass burning WS BrC. Given that it remains unclear to which extent fuel type and burn
conditions affect the molecular weight distribution of BrC chromophores, the observed low contribution of low-MW BrC to total light absorptivity may either be due to their rapid photochemical removal in the atmosphere, as demonstrated from our laboratory experiments, or their low emission rate. In addition, the average contribution of high-MW chromophores to total BrC light absorptivity is lower compared to WS BrC, suggesting that some water-insoluble BrC are low-MW compounds.

Given that the estimated atmospheric lifetime of ambient BrC from bulk measurements was estimated to be between
15 – 19 hours, and that high-MW chromophores contributed on average 75-87% of total light absorptivity for MeOH and WS ambient BrC, the atmospheric lifetime of ambient high-MW MeOH and WS BrC is likely to be at least 15 – 19 hours as well, which is longer than the estimates from laboratory results for the WS BrC ($\tau \sim 11 \pm 2.3$ hours, Table S1). We spectulate that the difference in atmospheric lifetime may be due to several reasons. Firstly, the laboratory constrainted atmospheric lifetime represents high-MW WS BrC that were cloud processed, which may not apply for the ambient samples. Secondly, high-MW



WS BrC were continuously exposed to UV radiation in the laboratory, where this does not represent the dinural cycle of solar radiation ambient BrC is exposed to. This is particularly important for high-MW WS BrC emitted throughout the night, where its atmospheric lifetime will be longer relative to those emitted during the day, during which the high-MW WS BrC are photobleached. Thirdly, the lifetime obtrained from laboratory results were based on the aging of BrC dissolved in bulk solutions, where parameters that may affect the reactivity of ambient BrC in suspended particles, such as aerosol phase state, solute concentrations, and viscosity were not accounted for. Similarly, ambient aeorsols containing biomass burning BrC are likely to be more chemically complex than those studied in the laboratory (e.g., complex emissions from the combustion of multiple types of biomass), where the presence of other organic compounds that are more reactive may prolong the atmospheric lifetime of ambient high-MW BrC. Lastly, there may be uncertinities associated with the estimated atmospheric transport times for field samples, either due to unaccounted contributions from a) fires occuring more than 3 days prior to filter collection (i.e., HYSPLIT runs times of 72 hours were used) or b) small fires with fire radiatve power less than 100 MW, which were not included in the current analysis, or c) large fires that are not reported by MODIS due to interference effects by thick smoke (Schroeder et al., 2008).

### 3.2.2 Chemical Tracers for Aged BrC from Biomass Burning

Previous studies have suggested the limitations of using levoglucosan and non-sea salt potassium (nss-$K^+$) as a chemical tracer for biomass burning, due to levoglucosan's short atmospheric lifetime (e.g., Hennigan et al., 2010; May et al., 2012) and emission of nss-$K^+$ by non-biomass burning sources (Urban et al., 2012). More recently, Scaramboni et al. (2015), suggested the use of total hydrous sugars (e.g., glucose) as an alternative tracer for biomass burning, since the six-membered ring of hydrated sugar molecule is potentially more stable than the five-membered ring of levoglucosan. While biological aerosols, such as microbes, are also a source of hydrated sugars (Graham et al., 2002) and may interfere with the signal from biomass burning, it is still useful to examine correlations with these types of sugars to BrC under known periods of biomass burning influence - especially given that high-MW BrC from biomass burning appears to be a relatively stable aerosol component, such that it can still be observed after significant periods of aging.

Focusing on the ambient filter samples impacted by biomass burning with estimated atmospheric transport times over 10 hours, we compare correlations of levoglucosan, nss-$K^+$, and total hydrous sugars (defined as the sum of glucose, mannose, and galactose mass concentration) to BrC, to assess how they compare to BrC as a biomass burning tracer for aged BBOA. Figure 7 shows their correlations to MeOH and WS BrC light absorption at 365 nm. The detailed results of these regression analysis are provided in Table S2. No correlation is seen between levoglucosan and aged MeOH and WS BrC ($r^2 \leq 0.01$), suggesting that levoglucosan is not likely to be representative of absorbing organic components, nor serve as an effective tracer of aged biomass burning. In comparison, both WS and total BrC correlated moderately with nss-$K^+$ ($0.50 \leq r^2 \leq 0.70$), consistent with previous field measurements of BBOA from the boreal forest (Di Lorenzo et al., 2018) and Amazon rainforest (Fuzzi et al., 2007). The moderate correlations may potentially be due to differences in the dependence of potassium emissions and BrC light absorptivity on fire conditions (Chen and Bond, 2010; Lee et al., 2010). Additionally, contributions of nss-$K^+$ from other non-biomass burning sources, such as crustal material, could also diminish the correlation with BrC if BrC is more specific to





biomass burning than nss-K$^+$. Moderate to strong correlations between light absorptivity of (MeOH and WS) BrC and total hydrous sugars were observed ($0.56 \leq r^2 \leq 0.83$), suggesting that both class of compounds may serve as a robust tracer for aged biomass burning emissions.

## 4. Conclusions and Atmospheric Implications

In this work, the effects of atmospheric aging on the light absorptivity of molecular weight separated-BrC were demonstrated in both controlled laboratory experiments and ambient observations. The laboratory work focused on the aging of WS BrC and the experimental conditions most likely represent cloud processing of BBOA. The ambient data included analysis of both WS and MeOH BrC. In the laboratory experiments, photochemical aging processes led to significant changes in light absorptivity and molecular weight distributions of BrC, where the reactivity of WS BrC was observed to be dependent on molecular weight. We found that low-MW WS BrC undergoes rapid photobleaching in the atmosphere on timescales of a few hours, whereas high-MW WS BrC likely persist in the atmopshere up to a few days. These laboratory results bridge constrasting results from previous laboratory and field observations that show low-MW WS BrC, such as nitrophenols, were rapidly photobleached within several hours, while high-MW WS BrC are more stable. Ambient BrC was largely composed of WS BrC species, and for both WS and MeOH BrC, their light absorptivity was dominated by high-MW BrC, consistent with our laboratory results. Ambient BrC was observed to undergo initial photoenhancement and high-MW BrC dominated total light absorption at 365nm, even for fresh (~1 hour of atmospheric transport) biomass burning emissions, further supporting that low-MW BrC are short lived components in atmospheric BBOA. Additionally, observations of initial photoenhancement due to atmospheric aging in both laboratory and field data support earlier ambient observations that secondary production of BrC from biomass burning emissions can be an important source of light absorbing aerosol (Gilardoni et al., 2016), but only near fire emissions. From the observed decay of bulk WS light absorptivity, we estimate that ambient WS BrC have atmospheric lifetimes of approximately 15 - 19 hours. This range of atmospheric lifetime likely corresponds to that of high-MW BrC, as they contributed over 75% to total light absorptivity of ambient BrC of ages up to 68 hours. It is important to note that while this estimated atmospheric lifetime likely represents a majority of high-MW BrC, the slope of the light absorpivity decay curve (Figure S6) decreased with atmospheric age, indicating that a fraction of the high-MW BrC are persistent in the atmosphere, for at least up to 68 hours. Collectively, this field-constrained lifetime of high-MW WS BrC is much larger than that obtained from laboratory results, which may be due to differences in the assumed versus ambient solar photon fluxes and oxidant concentrations. In addition, aging processes not investigated in the laboratory study, such as dilution, are also unlikely to result in more rapid removal of high-MW BrC from the atmosphere. Although the volatility of high-MW BrC remains uncharacterized, given their large molecular weights and solubility in water, we expect them to exhibit very low volatility and are unlikely to be lost from biomass burning particles via volatilization during dilution with background airmasses.

Given that the average lifetime of particles in the atmosphere is approximately one week with respect to deposition, the estimated atmospheric lifetime of high-MW BrC continue to support earlier observations that they can be persistent

©c Author(s) 2018. CC BY 4.0 License.



components in atmospheric BBOA (Di Lorenzo and Young, 2016; Wong et al., 2017; Di Lorenzo et al., 2017) and therefore have a larger impact on aerosol direct radiative forcing compared to low-MW BrC. The stability of high-MW BrC also suggests that they may be ubiquitous in the atmosphere and potentially undergo intercontinental transport. In addition, the estimated atmospheric lifetimes of ambient BrC from this study (15 – 19 hours) are consistent to those from previous field studies of

wildfires in Northwest USA (τ: 13 - 22 hours; Forrister et al., 2015) and in the Amazon (τ: 22 - 45 hours; Wang et al., 2016). A recent modelling study indicated that incorporating a 1-day photobleaching e-folding time, which was constrained from these previous field studies, improved modelled-vs-observed BrC absorption and decreased the estimated positive direct radiative effect of organic aerosols (Wang et al., 2018). Further field data from different geographical regions are necessary to assess the estimated 1-day BrC atmospheric lifetime and to improve predictions of global BrC impacts.

Accurate predictions of the direct radiative forcing also require robust estimates of the contribution of biomass burning to global BrC. Our field observations suggest total hydrous sugars and BrC may be more robust biomass burning markers compared to levoglucosan and nss-K$^+$. Other potential sources of hydrous sugars should also be assessed to better constrain their use as tracers for biomass burning.

From our field observations, the fractional contribution of high-MW MeOH and WS BrC to total (all molecular

weights) light absorptivity remained relatively constant up to 68 hours of atmospheric aging, suggesting that these class of compounds may also provide an alternative marker for aged biomass burning emissions. However, it remains unclear whether fuel type and burn conditions influence the emission, chemical composition, and reactivity of high-MW BrC. For example, while the absorptivity of MeOH and WS BrC generated from wood smoke has been shown to be dependent on pryolysis temperature and wood types (Chen and Bond, 2010), it remains unclear whether this is due to differences in the emisson of

low or high-MW BrC. Characterization of high-MW BrC generated from other commonly burned biomass, such as agriculture crop residues and biofuels used for residental heating, under representative combustion and fuel conditions (dry or wet), is warranted.

*Acknowledgements.* This work was supported by Electric Power Research Institute (EPRI) through contract #00-10003806

and NASA throguh constract NNX14A974G. Athanasios Nenes acknowledges support from the European Research Council Project PyroTRACH (Pyrogenic TRansformation Affecting Climate and Health) grant agreement 726165.

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

**Table 1. Rate constants for the photo-enhancement and photo-bleaching of low- and high-MW WS BrC of wood smoke, with respect to photolysis (UVB and UVA) and OH oxidation. Note that for UVB photolysis and OH oxidation, the reported uncertainties represent the variability ($\pm 1\sigma$) of multiple experiments (n=3) and that the rate constants for low-MW$_1$ and low-MW$_2$ BrC photobleaching corresponds chromophores that were rapidly and slowly photobleached, respectively. Rate constants for UVA photolysis were previously reported in Wong et al. (2017).**

|  | Fraction[a] | $k_{OH}^{II}$(M s$^{-1}$) | $k_{UVB}$(s$^{-1}$) | $k_{UVA}$(s$^{-1}$) |
|---|---|---|---|---|
| **BrC Photo-enhancement** | Low-MW | - | - | $(5.3 \pm 1.5) \times 10^{-5}$ |
|  | High-MW | - | $(1.2 \pm 0.2) \times 10^{-4}$ | $(9.2 \pm 1.4) \times 10^{-5}$ |
| **BrC Photo-Bleaching** | Low-MW$_1$ | - | $(3.5 \pm 0.7) \times 10^{-4}$ | $(1.8 \pm 0.4) \times 10^{-5}$ |
|  | Low-MW$_2$ | $(2.9 \pm 0.5) \times 10^{9}$ | $(1.6 \pm 0.3) \times 10^{-5}$ | |
|  | High-MW | - | $(1.7 \pm 0.4) \times 10^{-5}$ | $(1.5 \pm 0.6) \times 10^{-5}$ |

[a] High-MW BrC are defined as is defined as the sum of light absorptivity (Abs$_{365}$) for molecules with approximate molecular weights between 66K Da and 401 Da (i.e., SEC elution volumes between 8 and 15 mL) and low molecular weight fraction (low-MW) as the sum of light absorptivity for molecules with approximate molecular weights of 400 Da and less (i.e., SEC elution volumes higher than 15 mL).



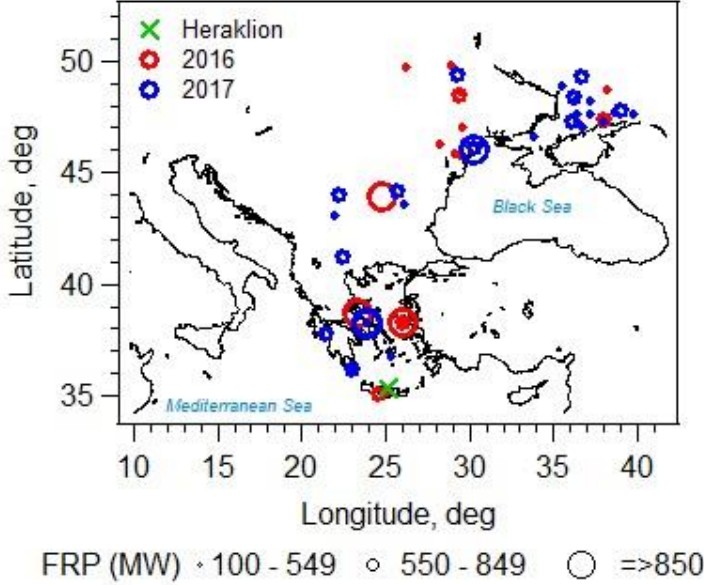

**Figure 1. Locations of biomass burning events relevant to the current study, as detected by the Moderate Resolution Imaging Spectroradiometer (MODIS)/the Fire Information for Resource Management System (FIRES) (http://firms.modaps.eosdis.nasa.gov/firemap) during the fire season of 2016 (red circles) and 2017 (blue circles). The corresponding MODIS-measured fire radiative power (in megawatts, FRP) is represented by the size of the circle markers, and is provided as a rough proxy of biomass burning emission rate (Wooster, 2002). Wildfires with FRP less than 100 MW are not shown here. The locations of the sampling site (Heraklion) is indicated by the green cross.**





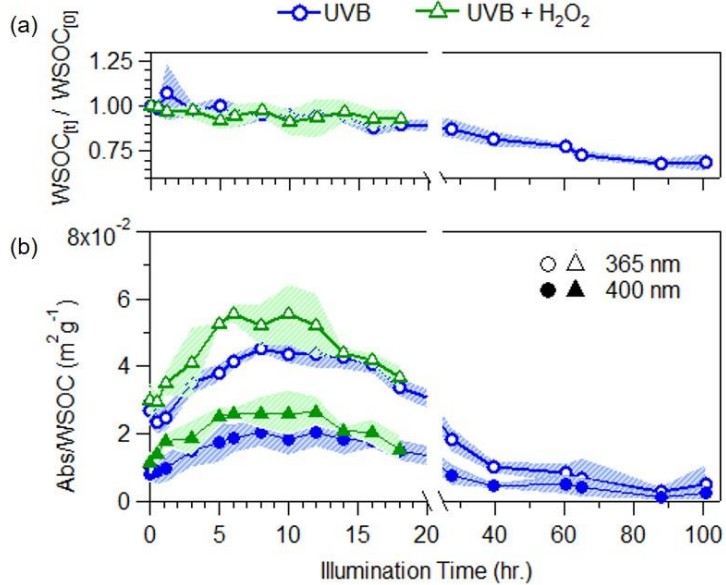

**Figure 2.** Effects of UVB photolysis (blue circles) and aqueous OH oxidation (green triangles) on: A) changes in WSOC, B) WSOC mass normalized absorption coefficient at 365 nm (open markers) and 400 nm (filled markers). The shaded areas represent the variability (±1σ) of multiple experiments (n=3).



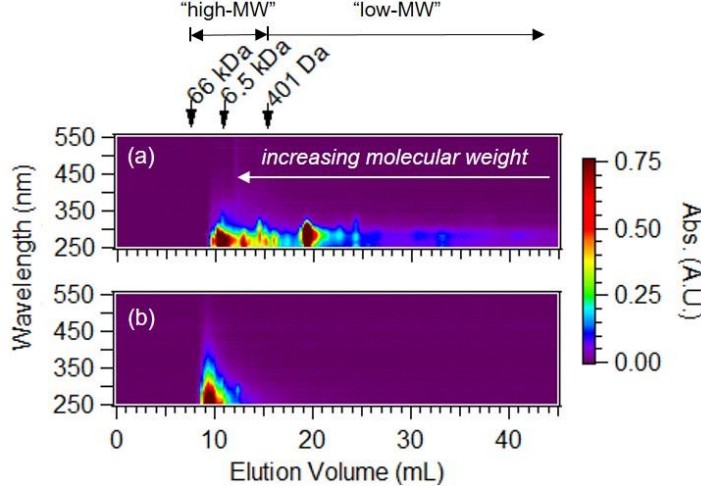

**Figure 3. Molecular weight separated absorption spectra of a) unreacted water soluble BrC from lab generated wood smoke and b) ambient WS BrC from fresh biomass burning emissions (~ 1 hour of atmospheric transport) collected during the 2016 fire season in Crete, Greece. The elution volumes for high-MW and low-BrC and some calibration standards with known molecular weights are provided for reference. Note that molecular weight increases with decreasing elution volumes.**





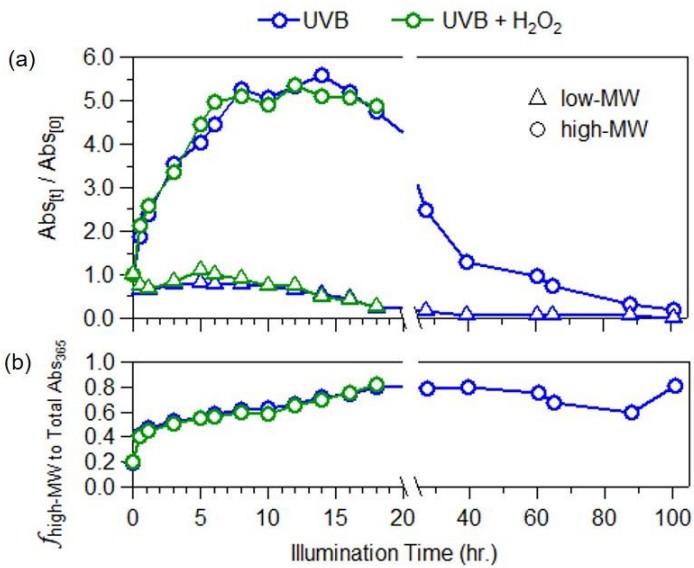

**Figure 4. Changes in a) light absorption at a wavelength of 365 nm for low-MW (triangles) and high-MW (circles) BrC fractions and b) contribution of high-MW fractions to total light absorbance at 365 nm in WS smoke BrC due to UVB photolysis (blue) and aqueous OH oxidation (green).**

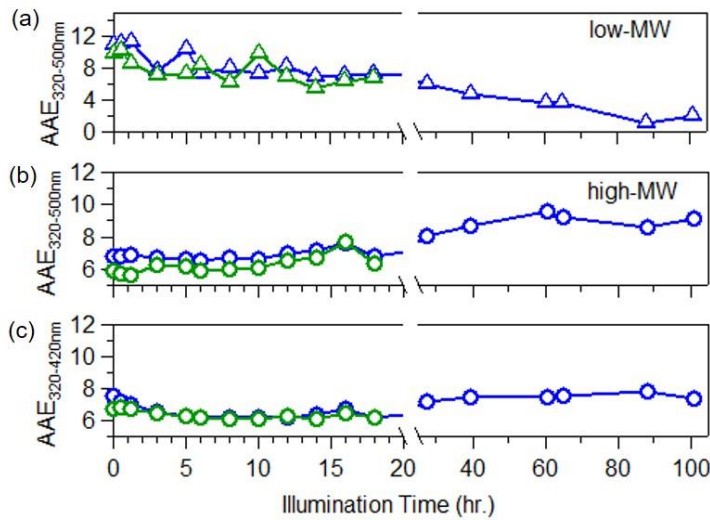

**Figure 5 Evolution of AAE due to UVB photolysis (blue) and aqueous OH oxidation (green) for a) low molecular weight BrC (triangles), b) high molecular weight BrC (circles) from 320-500 nm, and c) from 320-420nm for high molecular weight BrC. The error bars represent the variability ($\pm1\sigma$) of multiple experiments (n=3).**





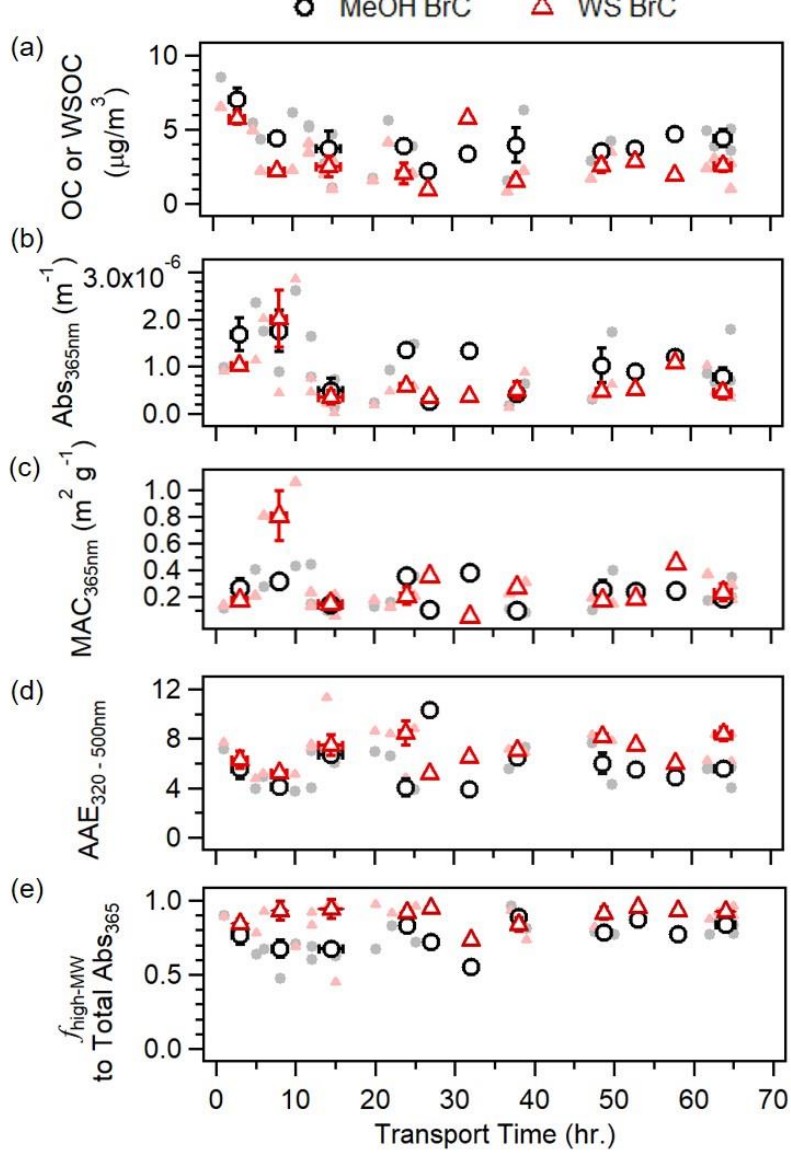

**Figure 6. a) OC or WSOC concentrations, b) light absorption at 365nm, c) mass absorption coefficients at 365nm, d) AAE (in the wavelength range of 320 - 500 nm) and e) fractional contribution of high-molecular weight fractions to total light absorption as a function of atmospheric transport time, for MeOH (black circles) and water (red triangles) extractable portion of the ambient filters collected on Crete Island, Greece during the 2016 and 2017 fire seasons. The filled points are individual filter measurement and the open points represent the binned median values and the associated error bars represent the interquartile range.**





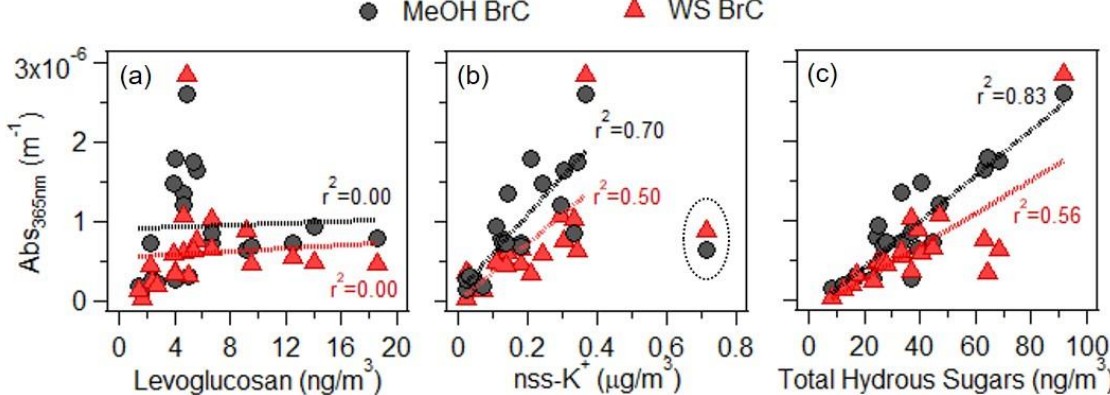

**Figure 7. Correlations of light absorption at 365 nm for the MeOH (black circles) and water-soluble (red triangles) extractable portion of ambient filter samples that were impacted by biomass burning to BB tracers: a) levoglucosan, b) nss-K⁺, and c) total hydrous sugars. For nss-K⁺, outliers (circled points) were excluded in the linear regression analysis.**

