# Peer review of "Atmospheric Evolution of Molecular Weight Separated Brown Carbon from Biomass Burning"

_Atmospheric Chemistry and Physics, 2018_

## Referee Comment (RC1) · Anonymous Referee #1 · 7 Oct 2018

This manuscript by Wong et al. investigates the evolution of water-soluble brown carbon (BrC) light absorption due to photolysis and OH oxidation for both laboratory generated BrC from the pyrolysis of cherry hardwood and ambient biomass-burning BrC collected in Crete. The results show interesting behavior starting with increase in absorption (photoenhancement) followed by decrease in absorption (photobleaching). The rates of these two stages were observed to depend on molecular weight, with larger molecules (> 400 Da) dominating the contribution to light absorption after several days of UV exposure. This is a well-written manuscript addressing a timely topic, and the findings constitute a step forward in the understanding of the evolution of atmospheric BrC. Therefore, I believe this manuscript is suitable for publication in ACP after the following comments are addressed:

1. It is not clear why both water and methanol extractions were performed for ambient samples and just water for laboratory samples. Please provide justification.

2. In section 2.1.1, it is stated that the BrC was produced by pyrolysis of cherry hardwood at 210 C and oxygen-free conditions. Is there a rationale behind choosing these conditions? It is known that the physicochemical properties and optical properties of BrC depend strongly on burn conditions (e.g. Chen and Bond (2010); Saleh et al. (2014); Pokhrel et al. (2016), etc.). While this is somehow acknowledged in the last sentence of the manuscript, the limitations of using one fuel and one burn condition should be featured early on when discussing the results (i.e. in section 3.1) as well as the abstract. This is not to take anything away from the interesting findings of this study.

3. Section 3.1.1: Describe how MAC is calculated. The reference to Wong et al. (2017) is not enough, as this is a central piece of this paper and it will benefit the reader to have at least a brief explanation of how MAC is calculated.

4. Also, you need a short discussion of the meaning of MAC in this context. I assume this is a "bulk" MAC, which is related to but not the same as the MAC of suspended particles. See, for example, discussion in section 3.1.2 of Laskin et al. (2015).

5. Finally, there is an inconsistency between the terminology in the text (MAC) and Figure 2 (AbsWSOC).

6. While the molecular-weight separated results are interesting, I believe further discussion is required. First, it would be helpful to include a more detailed description of the SEC technique. In particular, the dependence of MW on elution volume (from Figure 3) seems to be very non-linear. Please elaborate more on this and how it affects the measurements.

7. Second, what is the rationale behind a) grouping into just low and high MW (why not low, medium, and high, for example?) and b) choosing 400 Da as the cutoff?

8. Finally, Figure 3b shows that BrC absorption is mostly due to molecules larger than 10,000 Da. The observation of these large molecules, on its own, should be highlighted and discussed. Have such large molecules been observed in biomass burning BrC before?

References: Chen Y, Bond TC (2010) Light absorption by organic carbon from wood combustion. Atmos Chem Phys 10(2001):1773–1787. Saleh R, et al. (2014) Brownness of organics in aerosols from biomass burning linked to their black carbon content. Nat Geosci:DOI: 10.1038/NGEO2220. Pokhrel RP, et al. (2016) Parameterization of single-scattering albedo (SSA) and absorption Ångström exponent (AAE) with EC/OC for aerosol emissions from biomass burning. Atmos Chem Phys 16(15):9549–9561. Laskin A, Laskin J, Nizkorodov S a. (2015) Chemistry of Atmospheric Brown Carbon. Chem Rev:DOI: 10.1021/cr5006167.

---

## Referee Comment (RC2) · Anonymous Referee #2 · 8 Nov 2018

The authors present results from lab and field observations of absorption by brown carbon, generated from pyrolysis (lab) or from ambient fires (field). The lab experiments are well-done, although limited in scope given the focus on one source type (wood pyrolysis) that the authors have previously studied using similar methods (Wong et al., 2017, ES&T). Nonetheless, I believe the lab results are sufficiently novel and publishable after the authors address the reviewer comments. Regarding the field observations, I am much more skeptical of the results as they are currently presented and suggest they are removed unless further justification can be provided to provide a preponderance of evidence that they are correct for the reasons put forward.

In the intro (P2/L30 or so), the authors might consider also mentioning results from one of their earlier studies, Zhang et al. (2017, Nat. Geosci.).

P3/L15: It is not clear to me that the Zhou study observed "negligible concentrations of levoglucosan" in aged BBOA. They saw a small f60 in their AMS mass spectrum for the BBOA-3 factor. F60 is not the same as levoglucosan. It is also not completely clear, to me at least, that the Zhou BBOA-3 factor doesn't include secondary formation, given the extremely large O:C. The same comment applies to the Bougiatioti et al. study. Their BBOA factor is transformed over time to an "OOA" factor, but it is also not clear whether this completely excludes the influence of secondary formation: factors cannot tell one directly about the processes that led to their occurrence. The use of f60 is problematic simply because it can vary as a result of dilution by secondary material in addition to heterogeneous oxidation.

In describing their "biomass burning" in the abstract and introduction (P3, L18) I suggest that the authors identify this as "biomass burning particles produced from pyrolysis." It is, in my view, critical that the source of the BB particles be clearly identified for lab studies so that the reader, in the abstract, can begin to think about how this study compares to others. This is also important because the authors produce their BB particles under oxygen free conditions. It is not clear to me the extent to which this leads to the production of particles that are broadly representative of particles produced from open combustion, which includes oxygen. I suggest that the authors also comment on how representative they believe the particles produced from pyrolysis under oxygen-free conditions to be.

P4/L10: The authors note that most absorption comes from WSOC (77%). However, given that they find differnces in behavior between different WSOC fractions, the potential for the non-WSOC to behave differently than the WSOC seems large. Can the authors comment on this?

Figure S1: I think this would be better if the y-axis were a log scale, allowing the reader to compare at lower wavelengths better.

P8/L4: It would be helpful to the reader if the authors were to cmoment here on the

extent of the losses. While some loss is evident, it is also clear that over the first 20 h period the extent of loss is quite small (5%?). Also, the vials used for the experiments were sealed with Teflon-lined caps. Given this, where do the authors suspect the losses occurred? Did highly volatile vapors leave escape through the caps?

P8/L13: The authors state: "Given that a loss in WSOC was observed during this photoenhancement period, the increased MAC values may be driven by a loss in non-absorbing WSOC and/or the formation of more absorbing WS BrC." I find this argument to be relatively weak because the extent of increase (factors of 2) is substantially larger than the loss in WSOC mass (~5%).

P8/L14: The authors indicate that the MAC increase with H2O2 was simllar to that from UVB only. I agree that they are "similar" but it does seem that the changes in the H2O2 experiment was systematically larger. This is also connected with their observation of a steeper slope for H2O2 once things start to decline.

Abstract/L23: I suggest being more specific than "a few" hours. I would take this to mean around 3 h. Someone else might have a different number in mind. I suggest the authors just say over ~10 h (or whatever the right values are to insert). A numerical scale is given for the decay, so should be given for the rise as well. Also, the "rapid photobleaching" of the low-MW stuff doesn't seem to really capture the overall behavior, which has multiple components, including a slower component. I suggest that this is revised to more fully capture the behavior shown in Fig. 4

Data fitting: The authors extracted photoenhancement and photobleaching rate coefficients by fitting over different time periods. If the model proposed were robust, the authors should be able to describe the entire curve by fitting to both simultaneously. By fitting to separate regions, there is a strong assumption that there is no overlap in terms of the processes. Yet, they must overlap to obtain the flat-top region. I suggest it would be better if the authors were to perform a comprehensive fit across the entire data set to extract their parameters, showing the results of the fitting on the associated graphs.

[Figure]

This would help demonstrate the robustness of the model being used. Additionally, if the authors were to fit the observations using a more complete model they would not have to take a difference between the kUVB and kH2O2 to determine the OH-specific value. It would come naturally from fitting to the separate experiments. (Also, there appears to be a typo in Eqn. 2).

P10/L5: Given the many commas in this sentence, I find it somewhat difficult to follow.

P10/L29: It is not clear how this indicates that changes occurred "only up to 500 nm". More convincing would be an actual plot of the MAC at 500 nm, showing that there is no photoenhancement. But the use of the AAE difference seems to me a more indirect way of getting at this when a direct method is available.

Figures: On many, it would be helpful to the reader if tickmarks on the right sides were added.

P10/L27: The "slight decrease" in the high-MW AAE is not abundantly clear to me, especially for the OH experiment. There's a quick drop but then an increase and at 10 h the AAE seems even higher than at the start. Also, are the error bars alluded to in the caption of Fig. 5 smaller than the data points? The somewhat random behavior of the low-MW data suggests that error bars are lacking.

P11/L8: it is not clear to me how the authors are concluding that OH is the dominant loss mechanism. Fig. 4 shows that the low-MW stuff decays both from the UVB experient and the OH experiment to similar extents.

Field observations: I do not find the results presented especially compelling. There are two individual points for the WSOC MAC showing the increase. The data points around these two do not show evidence of any continuous evolution over time. Instead, there is a step function. And this step function is not seen with the MeOH MAC values. (Also, the increase in the MeOH absorption the authors mention at 10 h is not apparent to me. It seems to be flat within the data variability.) What can I take the uncertainty

as on these measurements? What should I make of the WSOC being substantially higher than the OC at the 33 h point in terms of uncertainty? What steps have the authors taken to convince themselves that this is not some artifact? Also, samples were collected for 22-24 h. Given this, what does it even mean to have an aging time of only 5 or 10 h? And were these daytime hours, given that there was sampling during day and night? If half the transport occurred at night, should the aging times be divided in two? Did the backtrajectories indicate that the air reaching the sampling site had been at the surface where the fire was in all cases? HYSPLIT gives altitude as well as location. Also, were these two samples for the same fires as at shorter or longer times? What if the nature of the fire and combustion conditions differed? Saleh et al. (2014, Nat. Geosci.) showed that the MAC (or the equivalent imaginary component) for the BrC varies with the combustion conditions. How can the authors rule out changes in conditions? I need to be further convinced about this analysis overall. Currently, I do not think it is sufficiently supported to be included in the manuscript.

P12/L18: I have a hard time believing that the AAE values are inversely proportional to the BC-to-OA based on the data presented. In my view, the observations show zero relationship, and the range of BC/OA is too small to show a relationship anyway. And, the authors are not accounting for the potential for secondary formation in the atmosphere, w hich would skew the relationship. I think this discussion should be struck as Fig. S7 does not show what the authors purport. I would need to be convinced with an R^2 at minimum.

The authors might consider commenting on the typical amount of time that BrC spends in a cloud droplet, since their results are most relevant to material in cloud droplets.

Levoglucosan: To what extent is it known whether the amount of levoglucosan produced relative to all other OA (or to BrC specifically) is a constant across varying burn conditions? This seems important to me to consider to understand the different relationships. The "total" hydrous sugars, I would speculate, is more likely to be consistent between burns than is a single sugar. As is something like K+, which would be relatively

independent of burn conditions (although would be fuel dependent). Nonetheless, the authors do convincingly show that the total sugars is more robust than levoglucosan alone.

It might be useful to know which two data points in Fig. 7 correspond to the two elevated MAC points in Fig. 6. It is not apparent than both of the elevated points are included in Fig. 7, because there should be two points with absorption > 2e-6 (from Fig. 6) but there is only one. The point that must be one of these is clearly one for which the relationship with levoglucosan is completely different than for any other points. Also for K+. This could indicate that the combustion conditions were, indeed, different for these high MAC points and thus that the elevated MAC was driven by differences in burn conditions and not aging.

There is no indication of where the data are archived, which I believe is now required by ACP.

---

## Author Comment (AC1) · 20 Dec 2018

**The authors thank the reviewer for their insightful comments and useful suggestions. We have provided a point-by-point response below.**

**Reviewer 1**

This manuscript by Wong et al. investigates the evolution of water-soluble brown carbon (BrC) light absorption due to photolysis and OH oxidation for both laboratory generated BrC from the pyrolysis of cherry hardwood and ambient biomass-burning BrC collected in Crete. The results show interesting behavior starting with increase in absorption (photoenhancement) followed by decrease in absorption (photobleaching). The rates of these two stages were observed to depend on molecular weight, with larger molecules (> 400 Da) dominating the contribution to light absorption after several days of UV exposure. This is a well-written manuscript addressing a timely topic, and the findings constitute a step forward in the understanding of the evolution of atmospheric BrC. Therefore, I believe this manuscript is suitable for publication in ACP after the following comments are addressed:

1. It is not clear why both water and methanol extractions were performed for ambient samples and just water for laboratory samples. Please provide justification.

**Response: We only focused on water extractions for laboratory sample since results from our previous work (Wong et al., 2017) indicated that the majority (77%) of the light absorption for laboratory generated BrC from wood smoke was contributed by the water-soluble fraction. This was stated in the original manuscript from lines 8-10. We also note that the trends in the evolution of light absorption for the methanol extraction for laboratory samples due to photochemical aging was also observed to be similar to the observations for water-soluble BrC fractions. We have modified section 2.1.1 (experimental section), as follows: "For these laboratory studies, we only focused on the aging of WS BrC, as results from our previous work indicated that the majority of the light absorption of laboratory generated BrC from wood smoke was contributed by the WS fraction (~ 77%) and that the trends in the evolution of light absorption of water-insoluble (i.e., methanol extracted) BrC due to photochemical aging are similar to that of the WS fraction (Wong et al., 2017)."**

2. In section 2.1.1, it is stated that the BrC was produced by pyrolysis of cherry hardwood at 210 C and oxygen-free conditions. Is there a rationale behind choosing these conditions? It is known that the physicochemical properties and optical properties of BrC depend strongly on burn conditions (e.g. Chen and Bond (2010); Saleh et al. (2014); Pokhrel et al. (2016), etc.). While this is somehow acknowledged in the last sentence of the manuscript, the limitations of using one fuel and one burn condition should be featured early on when discussing the results (i.e. in section 3.1) as well as the abstract. This is not to take anything away from the interesting findings of this study.

**Response: BrC was pyrolysed at 210°C under oxygen-free atmosphere to represent the smoldering conditions of wildfires, which we now have modified section 2.1.1 (experimental section) to mention this, as follows: "Wood smoke BrC was generated in the laboratory using the method described in Wong et al. (2017). Briefly, a small piece of dry cherry hardwood (5 - 10g), placed on the bottom of a cylindrical electronically-heated combustor, was pyrolyzed**

under an oxygen-free atmosphere at 210°C, to representing BrC emitted from smoldering combustion (Andreae and Gelencsér, 2006; Chen and Bond, 2010).”

We have also added the following statements in the specified sections of the manuscript to emphasize the use of one fuel type and burn conditions for the laboratory generated BrC:

Abstract: “Here we report laboratory experiments that examined changes in the optical properties of the water-soluble BrC fraction of laboratory generated biomass burning particles from hardwood pyrolysis.”

Section 3.2.1 (Results and discussion): “Firstly, the laboratory constrainted atmospheric lifetime represents BrC emitted from the combustion of one biomass fuel type under smoldering conditions, which may not represent ambient fire conditions, as the light absorption properties of BrC have been observed to be dependent on field and burn conditions (Chen and Bond, 2010).”

3. Section 3.1.1: Describe how MAC is calculated. The reference to Wong et al. (2017) is not enough, as this is a central piece of this paper and it will benefit the reader to have at least a brief explanation of how MAC is calculated.

Response: We have now described how the MAC values were calculated in section 2.1.4 (experimental section), as follows: “The measured light absorption for both bulk and molecular weight separated WS-BrC were normalized by the WSOC concentration of the BrC extract to represent the light absorption per water-soluble organic carbon, or mass absorption coefficient (MAC) of the WS-BrC.”

We have also added the following to section 2.2.2 to describe the calculation of MAC for WS and MeOH BrC for the ambient filter samples: “The MAC of the WS and MeOH soluble BrC were determined through normalizing by the WSOC concentration (WS BrC/WSOC) or OC concentrations (MeOH BrC/OC), as determined using the TOC and OCEC analyzers, respectively.”

4. Also, you need a short discussion of the meaning of MAC in this context. I assume this is a “bulk” MAC, which is related to but not the same as the MAC of suspended particles. See, for example, discussion in section 3.1.2 of Laskin et al. (2015).

Response: We have now added the following to section 3.1.1 to emphasize that the MAC represents the light absorption of the WS-BrC extract, “Note that these MAC values arise from light absorption measurements of water-extracted BrC and not of suspended BrC particles.”

5. Finally, there is an inconsistency between the terminology in the text (MAC) and Figure 2 (AbsWSOC).

Response: We have now modified the Figure 2 such that Abs/WSOC is now referred to MAC.

6. While the molecular-weight separated results are interesting, I believe further discussion is required. First, it would be helpful to include a more detailed description of the SEC technique. In particular, the dependence of MW on elution volume (from Figure 3) seems to be very non-linear. Please elaborate more on this and how it affects the measurements.

**Response: We agree that a discussion on the SEC technique would be helpful for readers, and have added a discussion of the SEC technique in section 2.1.4 (experimental section) as follows: "The molecular weight distributions of WS BrC were determined using size exclusion chromatography (SEC), which separates analyte molecules due to differences in the extent of permeation into the column packing material, where larger molecules elute earlier than smaller molecules due to weaker interactions (Strigel et al., 2009). The technique was operated using high-performance liquid chromatography (HPLC; GP40 Dionex), equipped with a SEC column (Polysep GFC P-3000, Phenomenex) that was operated in isocratic mode using a 90:10 *v/v* mixture of water and methanol with 25 mM ammonium acetate as the mobile phase, at 1mL min$^{-1}$."**

**We note that the elution time and molecular weight appears to be non-linear for Figure 3 because the relationship is linear for the logarithm of elution time to molecular weight. Since the elution of molecular weight separated BrC requires the use of a mobile phase, these dilution effects need to be account for in order to relate the absorbance of the molecular weight separated BrC to that of the bulk BrC sample. Since we have discussed these calculations in detail in our previous work (Wong et al., 2017), we have now added the following brief description in the section 2.1.4 (experimental section): "The absorbance of the different molecular weight fractions were determined by integrating the absorbance of a specific wavelength over the period of elution that corresponds to the molecular weight fraction ($P_{MW,\lambda}$). Since the coupling of the chromatographic technique to UV-VIS absorption measurements leads to the dilution of the BrC sample due to the use of the mobile phase, the absorbance of the molecular weight separated BrC is related to absorbance of the injected BrC by accounting for the mobile phase flow rate (*f*) and the volume of the injected BrC sample ($V_{BrC}$) using Eq. (1) :**

$$A_{BrC,\lambda} = \frac{P_{MW,\lambda} \cdot f}{V_{BrC}} \qquad (1)"$$

**New reference added: Strigel, Andre M., Yau, Wallace W., Kirkland, Joseph J. and Bly, Donald D.: Modern Size-Exclusion Liquid Chromatography, Second Edition., John Wiley & Sons, Inc., 2009.**

7. Second, what is the rationale behind a) grouping into just low and high MW (why not low, medium, and high, for example?) and b) choosing 400 Da as the cutoff?

**Response: The total light absorption was binned into two groups (high- and low-MW) instead of three because it is more reflective of the SEC separation process and better emphasize the uncertainties associated with the molecular weight information, as the accuracy of the SEC calibration approach, from which the MW information derives from, depends on whether the molecular densities of the calibration standards are representative of that of the BrC molecules. In other words, we feel the resolution of the SEC column combined with**

**uncertainties in calibration of the molecular weight separation does not warrant a more highly resolved classification then what we have used.**

**400 Da was designated as the cutoff for high- and low-MW BrC as it represents the penetration limit of the SEC column used.**

8. Finally, Figure 3b shows that BrC absorption is mostly due to molecules larger than 10,000 Da. The observation of these large molecules, on its own, should be highlighted and discussed. Have such large molecules been observed in biomass burning BrC before?

**Response: Ambient BrC molecules of molecular weights larger than ~10K Da have been previously observed in aged biomass burning aerosols collected at St. John's, Canada by De Lorenzo et al., 2017. We have highlighted this previous observation of high-MW BrC in the introduction and conclusion sections. We also note that given the large molecular weights and solubility in water, the high-MW BrC is likely to be of low-volatility, of which extremely low volatility organic compounds (ELVOCs) have been shown to dominate the light absorptivity of wood smoke BrC (Saleh et al., 2004). We have also added this point to the section 4 (conclusions and atmospheric implication).**

References:
Chen Y, Bond TC (2010) Light absorption by organic carbon from wood combustion. Atmos Chem Phys 10(2001):1773–1787.

Saleh R, et al. (2014) Brownness of organics in aerosols from biomass burning linked to their black carbon content. Nat Geosci:DOI: 10.1038/NGEO2220.

Pokhrel RP, et al. (2016) Parameterization of single-scattering albedo (SSA) and absorption Ångström exponent (AAE) with EC/OC for aerosol emissions from biomass burning. Atmos Chem Phys 16(15):9549–9561.

Laskin A, Laskin J, Nizkorodov S a. (2015) Chemistry of Atmospheric Brown Carbon. Chem Rev:DOI: 10.1021/cr5006167.

---

## Author Comment (AC2) · 20 Dec 2018

**The authors thank the reviewer for their insightful comments and useful suggestions. We have provided a point-by-point response below.**

**Reviewer 2**

The authors present results from lab and field observations of absorption by brown carbon, generated from pyrolysis (lab) or from ambient fires (field). The lab experiments are well-done, although limited in scope given the focus on one source type (wood pyrolysis) that the authors have previously studied using similar methods (Wong et al., 2017, ES&T). Nonetheless, I believe the lab results are sufficiently novel and publishable after the authors address the reviewer comments. Regarding the field observations, I am much more skeptical of the results as they are currently presented and suggest they are removed unless further justification can be provided to provide a preponderance of evidence that they are correct for the reasons put forward.

In the intro (P2/L30 or so), the authors might consider also mentioning results from one of their earlier studies, Zhang et al. (2017, Nat. Geosci.).

**Response: We have highlighted the results from our earlier work (Zhang et al., 2017) at the first paragraph of the introduction section in the original manuscript.**

P3/L15: It is not clear to me that the Zhou study observed "negligible concentrations of levoglucosan" in aged BBOA. They saw a small f60 in their AMS mass spectrum for the BBOA-3 factor. F60 is not the same as levoglucosan. It is also not completely clear, to me at least, that the Zhou BBOA-3 factor doesn't include secondary formation, given the extremely large O:C. The same comment applies to the Bougiatioti et al. study. Their BBOA factor is transformed over time to an "OOA" factor, but it is also not clear whether this completely excludes the influence of secondary formation: factors cannot tell one directly about the processes that led to their occurrence. The use of f60 is problematic simply because it can vary as a result of dilution by secondary material in addition to heterogeneous oxidation.

**Response: We agree with the reviewer that f60 is not the same as levoglucosan. We have now modified the sentence to clearly indicate that the studies of Zhou et al., and Bougiatioti et al., reported negligible mass spectrometric signatures of levoglucosan (i.e., f60), as follows: "Given that the average lifetime of atmospheric aerosols with respect to deposition is considerably longer, and that recent field observations have demonstrated aged BBOA had negligible concentrations of levoglucosan or of its mass spectrometric signatures (Bougiatioti et al., 2014; Zhou et al., 2017; Theodosi et al., 2018), these results suggest levoglucosan or mass spectral fragments cannot be used to estimate BrC levels in aged biomass burning emissions (> 1 day)."**

In describing their "biomass burning" in the abstract and introduction (P3, L18) I suggest that the authors identify this as "biomass burning particles produced from pyrolysis." It is, in my view, critical that the source of the BB particles be clearly identified for lab studies so that the reader, in the abstract, can begin to think about how this study compares to others. This is also important because the authors produce their BB particles under oxygen free conditions. It is not clear to me the extent to which this leads to the production of particles that are broadly representative of

particles produced from open combustion, which includes oxygen. I suggest that the authors also comment on how representative they believe the particles produced from pyrolysis under oxygen free conditions to be.

**Response: We agree that it should be highlighted in the abstract and introduction that the laboratory experiments were conducted on biomass burning particles emitted from biomass pyrolysis. The biomass burning particles, generated under oxygen-free condition, represents the smoldering phase of wildfires (Andreae and Gelencsér, 2006; Chen and Bond, 2010). We have now modified the manuscript to clarify these points (see response to Reviewer #1, question 2).**

**References:**
**Andreae, M. O. and Gelencsér, A.: Black carbon or brown carbon? The nature of light-absorbing carbonaceous aerosols, Atmos Chem Phys, 6(10), 3131–3148, doi:10.5194/acp-6-3131-2006, 2006.**

**Chen, Y. and Bond, T. C.: Light absorption by organic carbon from wood combustion, Atmos Chem Phys, 10(4), 1773–1787, doi:10.5194/acp-10-1773-2010, 2010.**

P4/L10: The authors note that most absorption comes from WSOC (77%). However, given that they find differences in behavior between different WSOC fractions, the potential for the non-WSOC to behave differently than the WSOC seems large. Can the authors comment on this?

**Response: In our previous work (Wong et al., 2017), we observed that the trends in the evolution of light absorption for the methanol extraction for laboratory samples due to photochemical aging by UVA aging was also observed to be similar to the observations for water-soluble BrC fractions. We have now added this point to the experimental section (please see response to Reviewer #1, question 1).**

Figure S1: I think this would be better if the y-axis were a log scale, allowing the reader to compare at lower wavelengths better.

**Response: Both UVA and UVB have similar photon fluxes at the smaller wavelengths and UVA lights have a much higher photon flux at longer wavelengths (as shown in Figure S1). As such, we believe that representing the axis for wavelength in a non-log scale is appropriate.**

P8/L4: It would be helpful to the reader if the authors were to comment here on the extent of the losses. While some loss is evident, it is also clear that over the first 20 h period the extent of loss is quite small (5%?). Also, the vials used for the experiments were sealed with Teflon-lined caps. Given this, where do the authors suspect the losses occurred? Did highly volatile vapors leave escape through the caps?

**Response: We have added and modified the discussion to mention the specific losses in WSOC that occur upon photochemical aging, as follows: "Upon illumination by UVB lights, losses in WSOC were observed for both direct UVB photolysis and UVB + $H_2O_2$ experiments (Figure 2a), with the majority of this loss occurring following 20 hours of illumination."**

**We can estimate the volume of the OC that partitioned into the gas-phase using the ideal-gas law: for a 30% loss of WSOC after 100 hours of illumination, with the WSOC concentration of the unreacted BrC solution (as mentioned in section 2.1.4 in the manuscript) at the room temperature and pressure, and assuming that the vapors have an average molecular weight of 200 g/mol, the vapors would have a volume (<0.1 uL), which is much less than the headspace of the glass vial. From this back-of-the-envelope calculation, the WSOC that was lost from the aqueous-phase (as highly volatile vapors) would have likely partitioned into the headspace of the 2mL sealed glass-vials (note that each vial originally contained 1 mL of the water extract of the laboratory generated biomass burning particles).**

P8/L13: The authors state: "Given that a loss in WSOC was observed during this photoenhancement period, the increased MAC values may be driven by a loss in nonabsorbing WSOC and/or the formation of more absorbing WS BrC." I find this argument to be relatively weak because the extent of increase (factors of 2) is substantially larger than the loss in WSOC mass (_5%).

**Response: This is a good point and should be highlighted in the discussion. We have now added the following discussion to section 3.1.1: "This initial increase in MAC values is likely driven by the formation of more absorbing WS BrC, as MAC values increased by a factor of ~ 2 while WSOC decreased only by factor of less than 1.1."**

P8/L14: The authors indicate that the MAC increase with H2O2 was similar to that from UVB only. I agree that they are "similar" but it does seem that the changes in the H2O2 experiment was systematically larger. This is also connected with their observation of a steeper slope for H2O2 once things start to decline.

**Response: We note that this part of the discussion pertains to the first 10 hours of the photochemical aging experiments. We have modified this sentence to clearly state that the changes in WSOC and MAC are similar for UVB and UVB + H₂O₂ experiments during this initial period of photochemical aging.**

Abstract/L23: I suggest being more specific than "a few" hours. I would take this to mean around 3 h. Someone else might have a different number in mind. I suggest the authors just say over _10 h (or whatever the right values are to insert). A numerical scale is given for the decay, so should be given for the rise as well. Also, the "rapid photobleaching" of the low-MW stuff doesn't seem to really capture the overall behavior, which has multiple components, including a slower component. I suggest that this is revised to more fully capture the behavior shown in Fig. 4

**Response: This is a good point. We agree that the timescale for the photoenhancement of high-MW BrC due to photochemical aging should be indicated. We have now modified the abstract to specify that the initial photoenhancement occurred up to ~ 15 hours. We have also specifically indicated that the majority of low-MW BrC undergo rapid photobleaching.**

Data fitting: The authors extracted photoenhancement and photobleaching rate coefficients by fitting over different time periods. If the model proposed were robust, the authors should be able to describe the entire curve by fitting to both simultaneously. By fitting to separate regions, there

is a strong assumption that there is no overlap in terms of the processes. Yet, they must overlap to obtain the flat-top region. I suggest it would be better if the authors were to perform a comprehensive fit across the entire data set to extract their parameters, showing the results of the fitting on the associated graphs. This would help demonstrate the robustness of the model being used. Additionally, if the authors were to fit the observations using a more complete model they would not have to take a difference between the kUVB and kH2O2 to determine the OH-specific value. It would come naturally from fitting to the separate experiments. (Also, there appears to be a typo in Eqn. 2).

**Response: Thank-you for pointing out the typo in Eqn. 2, which has been corrected. We also believe that the extracted rate constants from our current approach describes the *net* changes in light absorptivity due to photoenhancement and photobleaching processes, which can be directly compared to the field observations, which represent the net changes in light absorption due to either process. While we agree with that developing a more comprehensive kinetic model would extract more mechanistic information on the photoenhancement and photobleaching processes and should warrant further investigation. However, such a comprehensive kinetic model is not straight forward, two separate models will need to be develop, one for low-MW and one for high-MW BrC. Additionally, for the photoenhancement reaction, the relative light absorptivity between the original BrC reactant and the light absorbing product will need to be assumed – which is difficult given the fact that we do not know the concentration of the reactants vs. products (hence their molar-absorptivity). For these reasons, we decided not to use a more comprehensive model to estimate the photoenhancement and photobleaching rate constants.**

P10/L5: Given the many commas in this sentence, I find it somewhat difficult to follow.

**Response: We have now revised the sentence as follows: "For high-MW WS BrC, no differences in the evolution of their light absorption were observed between the two different photochemical aging experiments (Figure 4a). This suggests that their initial photoenhancement (up to 15 hours), and subsequent photobleaching, were only due to exposure to UVB radiation."**

P10/L29: It is not clear how this indicates that changes occurred "only up to 500 nm". More convincing would be an actual plot of the MAC at 500 nm, showing that there is no photoenhancement. But the use of the AAE difference seems to me a more indirect way of getting at this when a direct method is available.

**Response: We agree that differences in the changes in AAE values determined from different wavelength ranges can be used to assess spectral evolution of BrC due to aging, in addition to monitoring the evolution of MAC values at different wavelengths (approach that we demonstrated in Figure 2). We have now modified the statement to clearly indicate this, as follows: "Changes in the AAE values that were determined from different wavelength ranges can provide insight into the effects of photochemical aging on the light absorption spectral properties of BrC, in addition to monitoring the MAC values at different wavelengths."**

Figures: On many, it would be helpful to the reader if tickmarks on the right sides were added.

**Response: This is a great suggestion, we have now added tick marks to the right side of Figures 2, 4, 5, 6, and S6.**

P10/L27: The "slight decrease" in the high-MW AAE is not abundantly clear to me, especially for the OH experiment. There's a quick drop but then an increase and at 10 h the AAE seems even higher than at the start. Also, are the error bars alluded to in the caption of Fig. 5 smaller than the data points? The somewhat random behavior of the low-MW data suggests that error bars are lacking.

**Response: Thank-you for pointing this out. For Figure 5c, we have updated the scale for the y-axis such that the decrease in AAE$_{320\text{-}420nm}$ for high-MW WS BrC is clear. In addition, we now use shaded areas to present the experimental variability for Figure 5.**

P11/L8: it is not clear to me how the authors are concluding that OH is the dominant loss mechanism. Fig. 4 shows that the low-MW stuff decays both from the UVB experiment and the OH experiment to similar extents.

**Response: Aqueous OH oxidation was determined to be the dominant loss mechanism for the less reactive fraction of low-MW WS BrC as it resulted in the shortest estimated atmospheric lifetime compared to the other aging mechanisms studied in the laboratory (listed in Table S1).**

Field observations: I do not find the results presented especially compelling. There are two individual points for the WSOC MAC showing the increase. The data points around these two do not show evidence of any continuous evolution over time. Instead, there is a step function.

**Response: We agree that the evolution of WS BrC MAC appears to be a step function – a initial increase from 0 to 8 hours of atmospheric transport, followed by a decrease in MAC for WS BrC with atmospheric transport times of about 15 hours. We have discussed this trend in the original manuscript.**

And this step function is not seen with the MeOH MAC values. (Also, the increase in the MeOH absorption the authors mention at 10 h is not apparent to me. It seems to be flat within the data variability.)

**Response: We agree that the increase in MeOH BrC MAC values from 1 to 8 hours of atmospheric transport times is cannot be discern given the variability of the whole dataset. We have now explicitly stated this in section 3.2.1.**

What can I take the uncertainty as on these measurements?

**Response: The error bars for the filled points presents the interquartile range of the data points within a specific bin of atmospheric transport times.**

What should I make of the WSOC being substantially higher than the OC at the 33 h point in terms of uncertainty? What steps have the authors taken to convince themselves that this is not some artifact?

**Response: Thank-you for pointing this out. We have taken the following quality assurance and quality control measures for field samples:**
1) **multiple blanks were collected throughout the campaign to identify potential sampling contamination and limits of detection (LOD). The LODs (3 times the standard deviation of field blanks) were determined to be 0.30 µgC m$^{-3}$ for WSOC and 0.88 µgC m$^{-3}$ for OC, 1.7 x 10$^{-7}$ m$^{-1}$ for WS BrC light absorption, and 3.8 x 10$^{-7}$ m$^{-1}$ for MeOH light absorption. Only samples above LOD were used for further analysis.**
2) **more than half of the samples were analyzed in duplicates to assess reproducibility. Several filters samples from each fire season were also analyzed in triplet to quantify measurement variability. The measurement uncertainties were estimated to be 18% for WSOC/OC measurements, and 20% for the light absorption measurements (based on blanks, standards, and field samples duplicates).**

**Unfortunately, the WSOC measurement for the filter sample of atmospheric transport time of 33 hours was only conducted once and as pointed out, we should have noted that this sample had WSOC > OC. We have reanalyzed the filter in question in triplet and have clearly identified that the previous observation was an outlier, and that from this reanalysis work, OC > WSOC for this sample. We have also repeated this process for all samples were that previously only analyzed once (ten in total). We note that with the exception for the filter sample of atmospheric transport times of 33 hours, all the other repeated measurements were within the reported variability (~ 20%). We have now updated all the corresponding data and figures. In addition, we now mention the LOD and uncertainties of our measurements in the experimental section of the revised manuscript.**

Also, samples were collected for 22-24 h. Given this, what does it even mean to have an aging time of only 5 or 10 h? And were these daytime hours, given that there was sampling during day and night? If half the transport occurred at night, should the aging times be divided in two?

**Response: The denoted atmospheric transport time for each filter sample is the average atmospheric transport time as determined from each hourly trajectory that was computed during each filter sampling interval (i.e., the average time for the airmass to be transported from the biomass burning site to the sampling site). Given that the samples were collected for a period of 22 – 24 hours, the atmospheric transport time can include both day and nighttime hours. We note there are nightime aging processes that can transform the light absorptivity of BrC, such as dilution (McMeeking et al., 2014; Saleh et al., 2014).**

**References:**

**McMeeking, G. R., Fortner, E., Onasch, T. B., Taylor, J. W., Flynn, M., Coe, H. and Kreidenweis, S. M.: Impacts of nonrefractory material on light absorption by aerosols**

emitted from biomass burning, J. Geophys. Res. Atmospheres, 119(21), 12,272-12,286, doi:10.1002/2014JD021750, 2014.

Saleh, R., Robinson, E. S., Tkacik, D. S., Ahern, A. T., Liu, S., Aiken, A. C., Sullivan, R. C., Presto, A. A., Dubey, M. K., Yokelson, R. J., Donahue, N. M. and Robinson, A. L.: Brownness of organics in aerosols from biomass burning linked to their black carbon content, Nat. Geosci., 7(9), 647–650, doi:10.1038/ngeo2220, 2014.

Did the backtrajectories indicate that the air reaching the sampling site had been at the surface where the fire was in all cases? HYSPLIT gives altitude as well as location.

**Response: For filter samples that were influenced by biomass burning, all the corresponding back trajectories indicated that the airmasses were within 1000 m above ground level when they were at the biomass burning site. We note that the biomass burning plumes in Eastern Europe have been previously observed to have average injection heights of 3077 ± 951 m (Amiridis et al., 2010), so it is not necessarily the case that the airmass has to be at the surface to be influenced by the biomass burning emissions.**

References:

Amiridis, V., Giannakaki, E., Balis, D. S., Gerasopoulos, E., Pytharoulis, I., Zanis, P., Kazadzis, S., Melas, D. and Zerefos, C.: Smoke injection heights from agricultural burning in Eastern Europe as seen by CALIPSO, Atmospheric Chem. Phys., 10(23), 11567–11576, doi:https://doi.org/10.5194/acp-10-11567-2010, 2010.

Also, were these two samples for the same fires as at shorter or longer times? What if the nature of the fire and combustion conditions differed? Saleh et al. (2014, Nat. Geosci.) showed that the MAC (or the equivalent imaginary component) for the BrC varies with the combustion conditions. How can the authors rule out changes in conditions? I need to be further convinced about this analysis overall. Currently, I do not think it is sufficiently supported to be included in the manuscript.

**Response: For the samples that were included in the two MAC median values, they were all of different fires. We agree that the approach to identify biomass burning influence does not account for variability in BrC emissions from various fires, including duration, fire intensity, etc. We have stated these limitations in section 2.2.1 (experimental), as well as in section 3.2.1 (results and discussion). We also note that given the strong oxidizing environment of the Eastern Mediterranean, the field observations and their comparison to controlled laboratory photochemical experiments are valuable in evaluating the current understanding of the atmospheric evolution of BrC.**

P12/L18: I have a hard time believing that the AAE values are inversely proportional to the BC-to-OA based on the data presented. In my view, the observations show zero relationship, and the range of BC/OA is too small to show a relationship anyway. And, the authors are not accounting for the potential for secondary formation in the atmosphere, which would skew the relationship. I think this discussion should be struck as Fig. S7 does not show what the authors purport. I would need to be convinced with an $R^2$ at minimum.

**Response: It is a good point that from the current dataset, the relationship of AAE and BC-to-OA is difficult to discern, given the limited range of BC-to-OA ratios observed. We have removed Figure S7 and the associated discussion in the manuscript.**

The authors might consider commenting on the typical amount of time that BrC spends in a cloud droplet, since their results are most relevant to material in cloud droplets.

**Response: It is true that the BrC concentrations used in the laboratory experiment best represent those in cloud droplets. However, the aqueous aging chemistry investigated in this study can also occur in aqueous particles. We have noted in the original manuscript (section 3.2.1) that the laboratory work did not consider parameters that may affect the reactivity of ambient BrC in aqueous particles, such as viscosity, aerosol phase state, and solute concentrations.**

Levoglucosan: To what extent is it known whether the amount of levoglucosan produced relative to all other OA (or to BrC specifically) is a constant across varying burn conditions? This seems important to me to consider to understand the different relationships. The "total" hydrous sugars, I would speculate, is more likely to be consistent between burns than is a single sugar. As is something like K+, which would be relatively independent of burn conditions (although would be fuel dependent). Nonetheless, the authors do convincingly show that the total sugars is more robust than levoglucosan alone.

**Response: It is true that the emission of levoglucosan relative to OA and the relative emission of BrC to OA are not constant across varying burn conditions (Mazzoleni et al., 2007; Kalogridis et al., 2018) and could have led to the lack of correlation observed for BrC and levoglucosan. We have modified this section to include this point.**

**Added References:**

**Mazzoleni, L. R., Zielinska, B. and Moosmüller, H.: Emissions of Levoglucosan, Methoxy Phenols, and Organic Acids from Prescribed Burns, Laboratory Combustion of Wildland Fuels, and Residential Wood Combustion, Environ. Sci. Technol., 41(7), 2115–2122, doi:10.1021/es061702c, 2007.**

**Kalogridis, A.-C., Popovicheva, O. B., Engling, G., Diapouli, E., Kawamura, K., Tachibana, E., Ono, K., Kozlov, V. S. and Eleftheriadis, K.: Smoke aerosol chemistry and aging of Siberian biomass burning emissions in a large aerosol chamber, Atmos. Environ., 185, 15–28, doi:10.1016/j.atmosenv.2018.04.033, 2018.**

It might be useful to know which two data points in Fig. 7 correspond to the two elevated MAC points in Fig. 6. It is not apparent than both of the elevated points are included in Fig. 7, because there should be two points with absorption > 2e-6 (from Fig. 6) but there is only one. The point that must be one of these is clearly one for which the relationship with levoglucosan is completely different than for any other points. Also for K+. This could indicate that the combustion conditions were, indeed, different for these high MAC points and thus that the elevated MAC was driven by differences in burn conditions and not aging.

**Response: We note that only filter samples of atmospheric ages > 10 hrs (i.e., aged biomass burning emissions) were included in the analysis shown in Figure 7, as such, one of the filter samples with $Abs_{365} > 2$ e-6 m-1 were not included in Figure 7. It is true that the filter sample of high absorption corresponds to one of the elevate MAC values shown in Figure 6. It is a very good point that the relationship of $Abs_{365}$ to levoglucosan and potassium is clearly different for this high $Abs_{365}$ point compared to the rest of the data, which may suggest that the burning conditions for this filter sample is different.**

There is no indication of where the data are archived, which I believe is now required by ACP.

**Response: Thank-you for pointing this out. We have now provided all the data in a data repository. We have noted this in the section "Data Availability" at the end of the revised manuscript, before the acknowledge section.**